# MIXTURE OF STEP RETURNS IN BOOTSTRAPPED DQN

## ABSTRACT

The concept of utilizing multi-step returns for updating value functions has been adopted in deep reinforcement learning (DRL) for a number of years. Updating value functions with different backup lengths provides advantages in different aspects, including bias and variance of value estimates, convergence speed, and exploration behavior of the agent. Conventional methods such as TD ($\lambda$) leverage these advantages by using a target value equivalent to an exponential average of different step returns. Nevertheless, integrating step returns into a single target sacrifices the diversity of the advantages offered by different step return targets. To address this issue, we propose Mixture Bootstrapped DQN (MB-DQN) built on top of bootstrapped DQN, and uses different backup lengths for different bootstrapped heads. MB-DQN enables heterogeneity of the target values that is unavailable in approaches relying only on a single target value. As a result, it is able to maintain the advantages offered by different backup lengths. In this paper, we first discuss the motivational insights through a simple maze environment. In order to validate the effectiveness of MB-DQN, we perform experiments on the *Atari 2600* benchmark environments, and demonstrate the performance improvement of MB-DQN over a number of baseline methods. We further provide a set of ablation studies to examine the impacts of different design configurations of MB-DQN.

## 1 INTRODUCTION

In recent value-based deep reinforcement learning (DRL), a value function is usually utilized to evaluate state values, which stand for estimates of the expected long-term cumulative rewards that might be collected by an agent. In order to perform such an evaluation, a deep neural network (DNN) is employed by a number of contemporary value-based DRL methods (Mnih et al., 2015; Wang et al., 2016; Hasselt et al., 2016; Osband et al., 2016; Hessel et al., 2018) as the value function approximator, in which the network parameters are iteratively updated based on the agent's experience of interactions with an environment. For many of these methods (Mnih et al., 2015; Wang et al., 2016; Hasselt et al., 2016; Osband et al., 2016; Hessel et al., 2018), the update procedure is carried out by one-step temporal-difference (TD) learning (Sutton & Barto, 1998) (or simply "*one-step TD*"), which calculates the error between an estimated state value and a target differing by one timestep. One-step TD has been demonstrated effective in backing up immediate reward signals collected by an agent. Nevertheless, the long temporal horizon that the reward signals from farther states have to propagate through might lead to an extended learning period of the value function approximator.

Learning from multi-step returns (Sutton & Barto, 1998) is a way of propagating rewards newly observed by the agent faster to earlier visited states, and has been adopted in several previous works. Asynchronous advantage actor-critic (A3C) (Mnih et al., 2016) employs multi-step returns as targets to update the value functions of its asynchronous threads. Rainbow deep Q-network (Rainbow DQN) (Hessel et al., 2018) also utilizes multi-step returns during the backup procedure. The authors in (Barth-Maron et al., 2018) also modify the target value function of deep deterministic dolicy gradient (DDPG) (Lillicrap et al., 2016) to estimate TD errors using multi-step returns. Updating value functions with different backup lengths provides advantages in different aspects, including bias and variance of value estimates, convergence speed, and exploration behavior of the agent. Backing up reward signals through multi-step returns shifts the bias-variance tradeoff (Hessel et al., 2018). Therefore, backing up with different step return lengths (or simply '*backup length*' hereafter (Asis et al., 2018)) might lead to different target values in the Bellman equation, resulting in different exploration behaviors of the agent as well as different achievable performance of it. The authors

in (Amiranashvili et al., 2018) have demonstrated that the performance of the agent varies with different backup lengths, and showed that both very short and very long backup lengths could cause performance drops. These insights suggest that identifying the best backup length for an environment is not straightforward. In addition, although learning based on multi-step returns enhances the immediate sensitivity to future rewards, it is at the expense of greater variance which may cause the value function approximator to require more data samples to converge to the true expectation. Moreover, relying on a single target value with any specific backup length constrains the exploration behaviors of the agent, and might limit the achievable performance of it.

Based on the above observations, there have been several research works proposed to unify different target values with different backup lengths to leverages their respective advantages. The traditional TD ($\lambda$) (Sutton & Barto, 1998) uses a target value equivalent to an exponential average of all $n$-step returns (where $n$ is a natural number), providing a faster empirical convergence by interpolating between low-variance TD returns and low-bias Monte Carlo returns. DQN ($\lambda$) (Daley & Amato, 2019) further proposes an efficient implementation of TD ($\lambda$) for DRL by modifying the replay buffer memory such that $\lambda$-returns can be pre-computed. Although these methods benefit from combining multiple distinct backup lengths, they still rely on a single target value during the update procedure. Integrating step returns into a single target value, nevertheless, may sacrifice the diversity of the advantages provided by different step return targets.

As a result, in this paper, we propose **Mixture Bootstrapped DQN** (abbreviated as "*MB-DQN*") to address the above issues. MB-DQN is built on top of bootstrapped DQN (Osband et al., 2016), which contains multiple bootstrapped heads with randomly initialized weights to learn a set of value functions. MB-DQN leverages the advantages of different step return targets by assigning a distinct backup length to each bootstrapped head. Each bootstrapped head maintains its own target value derived from the assigned backup length during the update procedure. Since the backup lengths of the bootstrapped heads are distinct from each other, MB-DQN provides heterogeneity in the target values as well as diversified exploration behaviors of the agent that is unavailable in approaches relying only on a single target value. To validate the proposed concept, in our experiments, we first provide motivational insights on the influence of different configurations of backup lengths in a simple maze environment. We then evaluate the proposed MB-DQN on the *Atari 2600* (Bellemare et al., 2015) benchmark environments, and demonstrate its performance improvement over a number of baseline methods. We further provide a set of ablation studies to analyze the impacts of different design configurations of MB-DQN. In summary, the primary contributions of this paper include: (1) introducing an approach for maintaining the advantages from different backup lengths, (2) providing heterogeneity in the target values by utilizing multiple bootstrapped heads, and (3) enabling diversified exploration behaviors of the agent.

The remainder of this paper is organized as the following. Section 2 provides the background material related to this work. Section 3 walks through the proposed MB-DQN methodology. Section 4 reports the experimental results, and presents a set of the ablation analyses. Section 5 concludes this paper.

## 2 BACKGROUND

In this section, we provide the background material related to this work. We first introduce the basic concepts of the Markov Decision Process (MDP) and one-step return, followed by an explanation of the concept of multi-step returns. Next, we provide a brief overview of the Deep Q-Network (DQN).

### 2.1 MARKOV DECISION PROCESS AND ONE-STEP RETURN

In RL, an agent interacting with an environment $\mathcal{E}$ with state space $\mathcal{S}$ and action space $\mathcal{A}$ is often formulated as an MDP. At each timestep $t$, the agent perceives a state $s_t \in \mathcal{S}$, takes an action $a_t \in \mathcal{A}$ according to its policy $\pi(a|s)$, receives a reward $r_t \sim R(s_t, a_t)$, and transits to next state $s_{t+1} \sim p(s_{t+1}|s_t, a_t)$, where $R(s_t, a_t)$ and $p(s_{t+1}|s_t, a_t)$ are the reward function and transition probability function, respectively. The main objective of the agent is to learn an optimal policy $\pi^*(a|s)$ that maximizes discounted cumulative return $G_t = \sum_{i=t}^{T} \gamma^{i-t} r_t$, where $\gamma \in (0, 1]$ is the discount factor and $T$ is the horizon. For a given policy $\pi(a|s)$, the state value function $V^\pi$ and state-action value function $Q^\pi$ are defined as the expected discounted cumulative return $G_t$ starting

from a state $s$ and a state-action pair $(s, a)$ respectively, and can be represented as the following:

$$V^\pi(s) = \mathbb{E}[G_t | s_t = s, \pi], Q^\pi(s, a) = \mathbb{E}[G_t | s_t = s, a_t = a, \pi]. \tag{1}$$

In order to maximize $\mathbb{E}[G_t]$, conventional value-based RL methods often use one-step TD learning to iteratively update $V^\pi$ and $Q^\pi$. Take $Q^\pi$ for example, the update rule is expressed as the following:

$$Q(s_t, a_t) \leftarrow Q(s_t, a_t) + \alpha[r_t + \gamma Q(s_{t+1}, a_{t+1}) - Q(s_t, a_t)], \tag{2}$$

where $\alpha \in (0, 1]$ is a step size parameter which controls the update speed. This update procedure only considers the immediate return $r_t$ and $\gamma Q(s_{t+1}, a_{t+1})$, which is together called one-step return.

## 2.2 MULTI-STEP RETURN

Multi-step return is a variant of one-step return presented in the previous section. Multi-step return modifies the target of one-step return through bootstrapping over longer time intervals. It replaces the single reward $r_t$ in Eq. (2) with the truncated multi-step return $R_t^n$, which is represented as follows:

$$R_t^n = \sum_{j=0}^{n} \gamma^j r_{t+j}, \tag{3}$$

where $n$ is the selected backup length. The update rule of Eq. (2) is then re-written as the following:

$$Q(s_t, a_t) \leftarrow Q(s_t, a_t) + \alpha[R_t^n + \gamma^{n+1} Q(s_{t+n+1}, a_{t+n+1}) - Q(s_t, a_t)]. \tag{4}$$

A longer backup length $n$ has been shown to increase the variance of the estimated $Q(s_t, a_t)$ as well as decrease its bias (Jaakkola et al., 1994). Despite of the high-variance and the increased computational cost, multi-step return enhances the immediate sensitivity of the value approximator to future rewards, and allows them to backup faster. As a result, in certain cases, it is possible to achieve a faster learning speed for the value approximator by using an appropriate backup length $n$ larger than one (Sutton & Barto, 1998; Hessel et al., 2018; Amiranashvili et al., 2018).

## 2.3 DEEP Q-NETWORK

DQN is a DNN parameterized by $\theta$ for approximating the optimal Q-function. DQN is trained using samples drawn from an experience replay buffer $Z$, and is updated based on one-step TD learning with an objective to minimize a loss function $L_{DQN}$, which is typically expressed as the following:

$$L_{DQN} = \mathbb{E}_{s,a,r,s' \sim U(Z)}\big[(y_{s,a} - Q(s, a, \theta))^2\big], \tag{5}$$

where $y_{s,a} = r_t + \gamma \max_a Q(s_{t+1}, a, \theta^-)$ is the one-step target value, $U(Z)$ is a uniform distribution over $Z$, and $\theta^-$ is the parameters of the target network. $\theta^-$ is updated by $\theta$ at predefined intervals.

## 3 METHODOLOGY

In this section, we first demonstrate the impacts of different backup lengths on the behaviors of an agent in a simple maze environment. Then, we walk through the details of the MB-DQN framework.

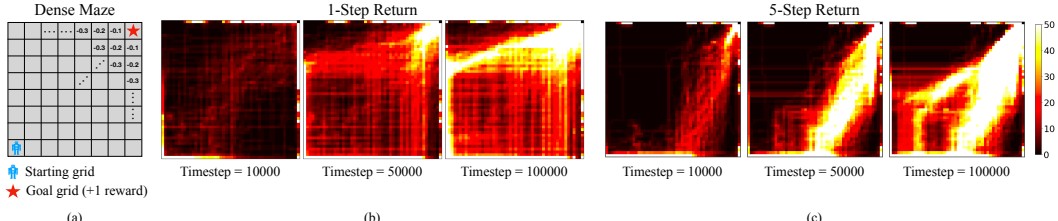

Figure 1: A visualization of the behaviors of the agents with different backup lengths. (a) presents the layout of the maze environment (denoted as *Dense Maze*), which contains a starting grid and a goal grid. (b) and (c) illustrate the behaviors of the agents updated using 1-step return and 5-step return, respectively. It is observed that the agent trained with 5-step return reaches the goal through shorter diagonal trajectories, while the agent trained with 1-step return explores more grids in the early stage.

### 3.1 AGENT BEHAVIOR WITH DIFFERENT BACKUP LENGTHS IN DQN

To illustrate the impacts of different backup lengths on an agent's behavior, we first consider a toy model in a two-dimensional maze environment containing a starting point and a goal, as depicted in Fig. 1 (a). We use DQN as our default agent and perform our experiments on this maze environment (denoted as *Dense Maze*) with dense rewards. In this setting, the reward of a grid gradually decreases as its distance to the goal increases. We depict the states visited by the agents for 100k timesteps in the training phase in Figs. 1 (b) and (c). It is observed that the agent trained with 5-step return reaches the goal through a shorter path than that of the 1-step return case. This is because the longer backup length allows the agent to adjust its value function estimation faster. On the other hand, although the agent trained with 1-step return might converge slower than that of the 5-step return case, it is observed that 1-step return enables the agent to visit and explore more states in the early stage. This is because the reward signal from a farther state has to propagate through a longer temporal horizon. Therefore, the agent explores more extensively before learning an effective policy to reach the goal.

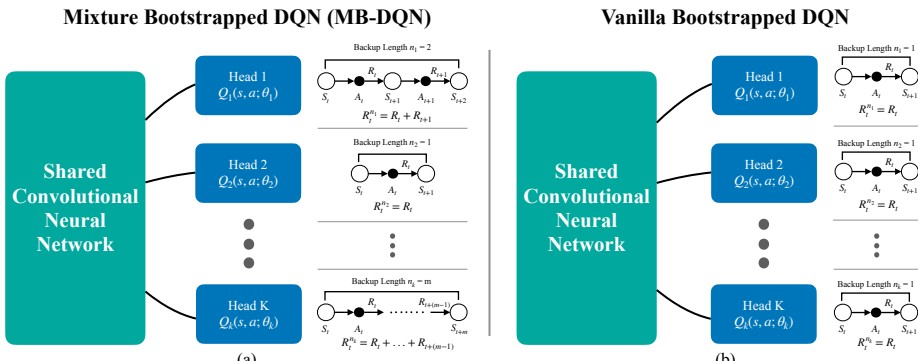

Figure 2: Overview of the proposed MB-DQN framework.

### 3.2 MIXTURE OF STEP RETURNS IN BOOTSTRAPPED DQN

In order to combine step returns with different backup lengths, we choose bootstrapped DQN (Osband et al., 2016) as our backbone framework. Bootstrapped DQN modifies DQN to approximate a distribution over Q-values via bootstrapping, and has demonstrated both the improved learning speed as well as the performance of the agents in various environments. At the beginning of an episode, bootstrapped DQN uniformly samples a Q-value function head $Q_k(s, a; \theta_k)$, $k \in \{1, ..., K\}$ from its $K$ bootstrapped Q-value function heads, as shown in Fig. 2. The agent then performs its control according to $Q_k(s, a; \theta_k)$ during the entire episode. Bootstrapped DQN re-samples a Q-value function head for each episode based on the same backup length (i.e., 1-step return) to calculate the target value (i.e., Eq. (2)). The framework, nevertheless, might be lack in diversity and heterogeneity among the bootstrapped heads.

As a result, we leverage the advantages of distinct Q-value function heads in bootstrapped DQN, and propose the usage of mixture backup lengths for different bootstrapped Q-value function heads in our MB-DQN framework, which is shown in Fig. 2. MB-DQN is similarly implemented as $K$ bootstrapped heads for estimating the Q-value function, where each bootstrapped head $k \in K$ correspond to its own backup lengths $n_k$. In each episode, MB-DQN also uniformly and randomly selects a head $k \in \{1, ..., K\}$, and stores the state transition data collected by the agent using this head into a replay buffer. The replay buffer is played back periodically to update the parameters of all the bootstrapped Q-value function heads as well as the shared convolutional neural network. Each head is trained with its own target network $Q_k(s, a; \theta_k^-)$ and its own target value $y_{s,a}^k$ with the truncated multi-step return defined in Eq. (3). The detailed update method is summarized in Algorithm 1, while the training methodology is the same as bootstrapped DQN (Osband et al., 2016). The truncated multi-step returns with different backup lengths thus provide diversity and heterogeneity for the $K$ bootstrapped estimates, which balance the strengths and the weaknesses of different backup lengths.

---

**Algorithm 1** Update Methodology of MB-DQN

---

1: Initialize K Q-networks $Q_k(s, a; \theta_k)$ with random weights $\theta_k$
2: Let each networks $Q_k$ with its own backup length $n_k$
3: **for** each update time **do**
4:     **for** each head k = 1, 2,...,K **do**
5:         $R_t^{n_k} = \sum_{j=0}^{n_k} \gamma^j R_{t+j}$
6:         $y_{s,a}^k = R_t^{n_k} + \gamma^{n_k} \max_a Q_k(s_{t+n_k}, \arg\max_a Q(s_{t+n_k}, a; \theta_k); \theta_k^-)$
7:         $\theta_k \approx \arg\min_{\theta_k} \mathbf{E}(y_{s,a}^k - Q_k(s, a; \theta_k))$

---

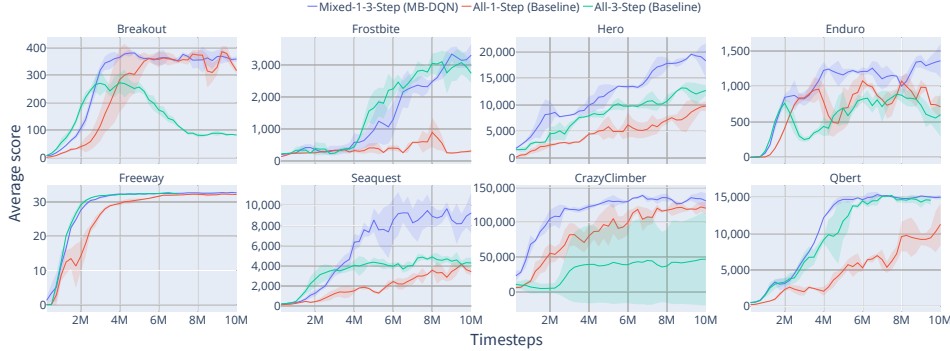

Figure 3: Comparison of the evaluation curves of MB-DQN and the baselines in eight *Atari* games. [1]

# 4 EXPERIMENTAL RESULTS

In this section, we present the experimental results to demonstrate the advantages of the mixture usage of different backup lengths. We first evaluate the proposed MB-DQN on a collection of well-known *Atari 2600* (Bellemare et al., 2015) games, and compare its performance to different configurations of boostrapped DQN both quantitatively and qualitatively in Section 4.1. Next, we investigate the quality of the data samples collected by MB-DQN, and demonstrate their advantages in training an RL agent in Section 4.2. Then, we validate the assumption made in Section 1 that unifying different step return targets to a single target value may not be as effective as the proposed MB-DQN approach in Section 4.3. Finally, we further provide a set of ablation analyses of MB-DQN on *Atari* games to inspect and discuss the impacts of different configurations on MB-DQN's performance in Section 4.4.

## 4.1 COMPARISON OF MB-DQN AND BOOTSTRAPPED DQN

**Environments.** To demonstrate the advantages of the mixture usage of different backup lengths, we begin with a collection of well-known *Atari* games, and plot the evaluation curves of eight common *Atari* games selected from the four different categories (Bellemare et al., 2016), including *Breakout*, *Frostbite*, *Hero*, *Enduro*, *Qbert*, *Seaquest*, *CrazyClimber*, and *Freeway*. We additionally provide the experimental results of the remaining games (33 *Atari* games considered in total) in the appendix.

**Baselines.** We compare the proposed MB-DQN against two baselines: bootstrapped DQN with (a) all 1-step return heads and (b) all 3-step return heads, which are denoted as *All-1-Step (Baseline)* and *All-3-Step (Baseline)*, respectively. We use ten bootstrapped heads for both MB-DQN and the baselines. MB-DQN (denoted as *Mixed-1-3-Step (MB-DQN)*) is implemented using five bootstrapped heads with 1-step backup length and another five bootstrapped heads with 3-step backup lengths.

**Quantitative comparison.** The qualitative comparison is presented in Fig. 3. It is observed that longer backup lengths do not always guarantee better performances — each environment may have its own favor. The *All-3-Step* baseline outperforms the *All-1-Step* baseline in six out of eight games,

---

[1]For all of our experiments, the MB-DQN and the bootstrapped DQN agents are evaluated every 250k timesteps based on the results voted by the majority of their bootstrapped heads. The evaluation curves are averaged from three random seeds, and are drawn with 68% confidence interval, illustrated as the shaded regions.

while it underperforms the *All-1-Step* baseline in the rest two games and suffers from a considerable performance drop in *Breakout*. In contrast, the proposed MB-DQN that uses a mixture of step returns outperforms the baselines in terms of its performance and convergence speed for most of the cases.

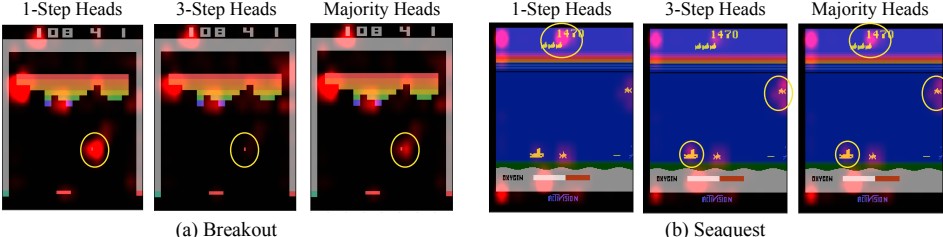

|  | 1-Step Heads | 3-Step Heads | Majority Heads |  | 1-Step Heads | 3-Step Heads | Majority Heads |

(a) Breakout                                        (b) Seaquest

Figure 4: Visualization of the agents' attention areas (rendered in red). The attention areas are derived based on (Greydanus et al., 2018) and are obtained from the bootstrapped heads of MB-DQN.

**Qualitative comparison via attention maps.**    In order to understand the rationale behind the high performance and advantages offered by MB-DQN, we further visualize the attention areas (Greydanus et al., 2018) of the agents trained with MB-DQN for three cases: attention areas generated from (a) all of the 1-step bootstrapped heads in MB-DQN (denoted as *1-Step Heads*), (b) all of the 3-step bootstrapped heads in MB-DQN (denoted as *3-Step Heads*), and (c) the composition of the bootstrapped heads which contribute to the decided actions (i.e., the majority of the bootstrapped heads that vote the resultant actions, denoted as *Majority Heads*). Please note that the composition in (c) may contain both the 1-step and 3-step bootstrapped heads. Fig. 4 illustrates the attention areas (rendered in red) of these three cases for two *Atari* games: *Breakout* and *Seaquest*, and highlights their differences by yellow circles. In *Breakout*, it is observed that the *1-Step Heads* case focuses more on the ball, the most important object in this game, than the *3-Step Heads* case. It is also observed that when the majority of the bootstrapped heads is considered, the attention of the agent is fell on the ball as well, allowing MB-DQN to play as good as the *All-1-Step* baseline in *Breakout* in Fig. 3. In *Seaquest*, it is observed that the *1-Step Heads* case focuses more on the scoreboard, while the *3-Step Heads* case focuses more on the enemy and the submarine. The attention areas of the *Majority Heads* case, on the other hand, cover the areas from both the *1-Step Heads* and the *3-Step Heads* cases, allowing MB-DQN to outperform the two baselines in Fig. 3. These examples therefore validate that MB-DQN is able to leverage the advantages from different backup lengths, and can achieve superior performance to the two baselines by offering heterogeneity among its bootstrapped heads.

## 4.2    ANALYSIS OF THE DATA SAMPLE QUALITY FOR MB-DQN

As the experimental results presented in the previous section have quantitatively and qualitatively demonstrated the performance benefits offered by MB-DQN, we next dive further to investigate the rationale behind the advantages. We hypothesize that the performance improvements provided by MB-DQN may come from the quality of the collected data samples in the experience replay buffer. In other words, MB-DQN may have benefited from the heterogeneity in the data samples collected by bootstrapped heads with different backup lengths. To validate this hypothesis, we design an experiment containing two agents: one agent is responsible for generating state-action pairs for an experience replay buffer while updating its Q-value network with the data contained in it. The other agent only updates its Q-value network by the existing data samples contained in the replay buffer, without contributing data to it. Both of these agents are implemented with ten bootstrapped heads.

We consider three configurations for the former *data generation agent*: *Mixed-1-3-Step (MB-DQN)*, *All-1-Step (Baseline)*, and *All-3-Step (Baseline)*, and two configurations for the latter *learning-only agent* (i.e., the one without contributing data samples to the replay buffer): *All-1-Step (Baseline)* and *All-3-Step (Baseline)*. The configurations are evaluated on *Seaquest*, and are designed such that the former and the latter agents have different configurations. The results of our experiments are plotted in Fig. 5 (a), where the figures on the top and bottom sides correspond to the cases that the *learning-only agents* are configured to *All-1-Step (Baseline)* and *All-3-Step (Baseline)*, respectively. It can be observed that the *learning-only agents* trained with the data samples generated by *Mixed-1-3-Step (MB-DQN)* outperform the agents trained with the data samples generated by the other

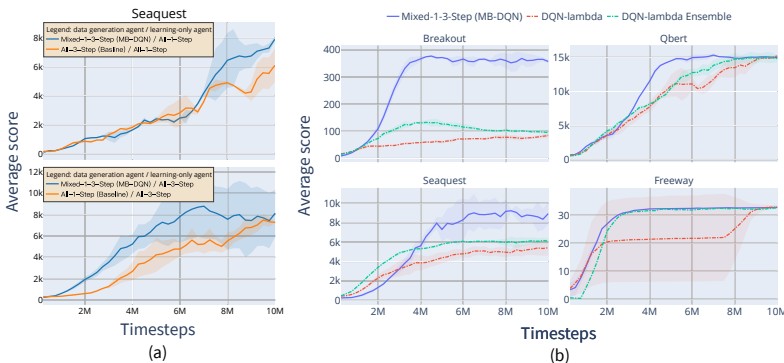

Figure 5: The evaluation curves of (a) comparison of different configurations of *data generation agents* and *learning-only agents* for validating the quality of data samples collected by MB-DQN in Section 4.2, and (b) comparison between the single $\lambda$-target strategy adopted by DQN ($\lambda$) (Daley & Amato, 2019) and the multiple bootstrapped targets strategy adopted by MB-DQN in Section 4.3.

configurations in both cases. These results thus validate our hypothesis that the data samples generated by MB-DQN are superior in quality than those generated by the other configurations, and explain why MB-DQN is able to offer benefits in performance in the environments presented in this paper.

### 4.3 $\lambda$-TARGET VERSUS HETEROGENEOUS BOOTSTRAPPED TARGETS

In order to validate our assumption in Section 1 that unifying different step return targets to a single target value may not be as effective as the heterogeneous bootstrapped approach adopted by MB-DQN, in this section, we compare these strategies of combining step returns in several *Atari* environments. For the unified return target strategy, we consider a recently proposed method called *DQN ($\lambda$)* (Daley & Amato, 2019), which implements TD ($\lambda$) by pre-computing $\lambda$-returns using an additional cache for its replay buffer memory. On the other hand, MB-DQN employs a strategy that leverages $K$ heterogeneous bootstrapped heads, where each head $k \in K$ has its own target value. For a fair comparison, we further include a variant of DQN ($\lambda$), called *DQN ($\lambda$) Ensemble*, which employs $K$ bootstrapped heads using $\lambda$-return as the target value. In our experiments, We set $K = 10$ for MB-DQN and DQN ($\lambda$) Ensemble, where the settings for DQN ($\lambda$) are configured as its default values (Daley & Amato, 2019). The single target value used by DQN ($\lambda$) is derived from multiple backup lengths ranging from one to a hundred. The evaluation curves of these strategies are plotted in Fig. 5 (b). It can be observed that for the four environments presented in Fig. 5 (b), the curves corresponding to the heterogeneous bootstrapped targets strategy (i.e., MB-DQN) grow faster and higher than those corresponding to the single-unified target strategy (i.e., DQN ($\lambda$) and DQN ($\lambda$) Ensemble). The above interesting evidence not only validates our assumption in Section 1, but also reveals that the advantages offered by the heterogeneity in multiple target values may outweigh the advantages offered by a single TD ($\lambda$) target that aggregates returns from the long temporal horizon.

### 4.4 ABLATION ANALYSIS

In this section, we provide a set of ablation analyses for the proposed MB-DQN on four selceted *Atari* games, including *Breakout*, *Qbert*, *Seaquest*, and *Freeway*, to examine the impacts of different configurations on MB-DQN's performances. We perform two sets of analyses for MB-DQN: (a) different configurations of step returns for the bootstrapped heads, and then (b) different numbers of the bootstrapped heads. Please note that additional experimental results are provided in the appendix.

#### 4.4.1 DIFFERENT CONFIGURATIONS OF STEP RETURNS FOR THE BOOTSTRAPPED HEADS

We consider various step returns less than or equal to three, and analyze four different configurations of step returns for the bootstrapped heads in MB-DQN, including *Mixed-1-2-3-Step*, *Mixed-1-2-Step*, *Mixed-1-3-Step*, *Mixed-2-3-Step*, and *All-1-Step (Baseline)*. All of these configurations contain ten bootstrapped heads. The configuration *Mixed-1-2-3-Step* consists of three 1-step bootstrapped heads, three 2-step bootstrapped heads, and four 3-step bootstrapped heads. For the configurations *Mixed-*

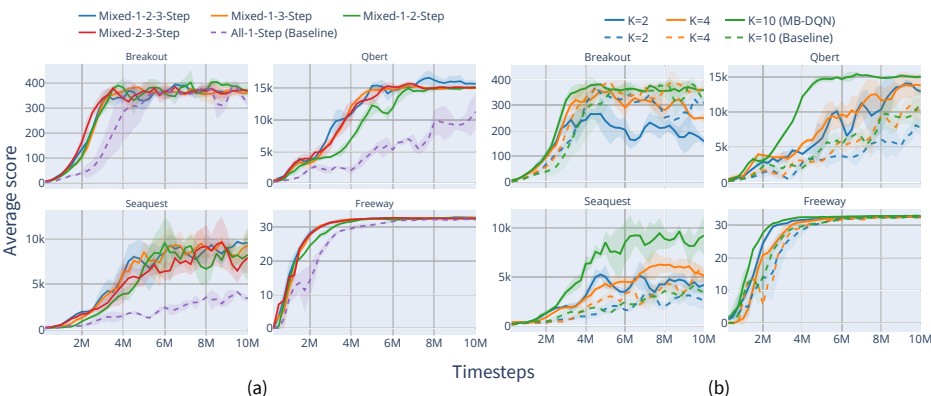

Figure 6: Impacts of (a) different configurations of step returns for the bootstrapped heads, and (b) different numbers of the bootstrapped heads on the proposed MB-DQN for four different *Atari* games.

*1-2-Step*, *Mixed-1-3-Step*, and *Mixed-2-3-Step*, the ten bootstrapped heads are evenly distributed to different backup lengths. The results are presented in Fig. 6 (a). For all of the configurations, it is observed that the agents trained with different mixtures of step returns perform similarly, and outperform those trained with the *All-1-Step* baseline. These evaluation results thus suggest that the bootstrapped heads in the proposed MB-DQN is not limited to certain configurations of step returns.

### 4.4.2 DIFFERENT NUMBERS OF THE BOOSTRAPPED HEADS

In bootstrapped DQN (Osband et al., 2016), more bootstrapped heads lead to faster learning, while even a small number of bootstrapped heads is still able to capture most of its benefits. As MB-DQN inherits its architecture from bootstrapped DQN, we investigate the impacts of different numbers of the bootstrapped heads on MB-DQN, and examine if MB-DQN still maintains this property or not. We perform a set of experiments with three different configurations of the bootstrapped heads: $K = 2, 4$, and 10. For each of the configuration, MB-DQN is set to consist of $K/2$ 1-step bootstrapped heads and $K/2$ 3-step bootstrapped heads. On the contrary, the bootstrapped DQN baseline is set as its default configuration and is implemented with $K$ 1-step bootstrapped heads.

Fig. 6 (b) illustrates the evaluation curves of the above configurations. In most cases, it is observed that for both MB-DQN and bootstrapped DQN, more bootstrapped heads lead to better performance. It is worth noticing that the proposed MB-DQN trained with two bootstrapped heads outperforms the baseline trained with ten bootstrapped heads in three of four games. This fact shows the significance and advantage of the mixture usage of multi-step returns in bootstrapped heads. On the other hand, MB-DQN's performance drops in *Breakout* as the number of the bootstrapped heads $K$ become less than that of the baseline. This is caused by the fact that the *3-Step Heads* perform worse than the *1-Step Heads* in *Breakout*, as described in Section 4.1. As a result, a smaller $K$ strengthens the negative influence caused by the *3-Step Heads*, causing MB-DQN to become sensitive to the undesirable performance in certain bootstrapped heads. The above results thus suggest that an appropriate number of $K$ has to be selected in order to maintain the advantages as well as the performance of MB-DQN.

## 5 CONCLUSION

In this paper, we proposed MB-DQN for combining and leveraging the advantages of different step return targets using multiple bootstrapped heads. Instead of unifying different step return targets to a single target value, MB-DQN assigns a distinct backup length to each bootstrapped head. This allows MB-DQN to offer heterogeneity in the target values during its update procedure, and enables a DRL agent to have diversified exploration behaviors. In our experiments, we first provided motivational examples to demonstrate the influence of different configurations of backup lengths in a simple maze environment. We then evaluated the proposed MB-DQN methodology on a number of *Atari 2600* environments both quantitatively and qualitatively, and validated that MB-DQN is able to outperform a number of baseline methods with different configurations of backup lengths. Finally, we presented a set of ablation studies to examine the impacts of different design configurations for MB-DQN.

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

# Appendices

## A1    ADDITIONAL BACKGROUND MATERIAL

In this section, we provide additional background materials related to our work. We first introduce the concept of DQN $(\lambda)$ (Daley & Amato, 2019), which is compared in Section 4.3. Next, we explain the generation method of the attention maps (Greydanus et al., 2018) used for the qualitative comparisons in Section 4.1 of the main manuscript.

### A1.1    DQN $(\lambda)$

DQN $(\lambda)$ (Daley & Amato, 2019) incorporates the concept of $\lambda$-return into DQN by modifying the replay buffer, with an aim to reduce the computation time required for deriving $\lambda$-return. The replay buffer is modified to store the $\lambda$-return $R_t^\lambda$ at timestep $t$ along with its corresponding transition. The value of $R_t^\lambda$ is computed in a recursive fashion, allowing repeated computations of $\lambda$-returns to be reduced when the same transitions are sampled multiple times. The recursive rule of $R_t^\lambda$ is thus expressed as follows:

$$R_t^\lambda = R_t^1 + \gamma\lambda[R_{t+1}^\lambda - \max_{a \in A} Q(s_{t+1}, a)], \tag{A1}$$

where $R_t^1 = r_t + \max_{a \in A} Q(s_{t+1}, a)$ is the one-step return target at timestep $t$, $Q$ is the state-action value function, $\gamma$ is the discount factor, and $s_{t+1}$ is the next state. In order to reuse outdated $\lambda$-returns caused by Q-function updates, DQN $(\lambda)$ introduces a mechanism which periodically samples random intervals of consecutive transitions from the experience replay buffer, and stores them into another cache memory. During each update, the transitions in the cache are refreshed by the current Q-function. The agent then samples batches of training data from the relatively smaller cache using a prioritized sampling mechanism similar to the one used by the prioritized replay buffer (Schaul et al., 2016) (i.e., the larger the TD error of the transition is, the more likely it would be sampled). DQN $(\lambda)$ has been validated with two different $\lambda$-return estimators, Peng's Q $(\lambda)$ (Peng & Williams, 1994) and Watkin's Q $(\lambda)$ (Watkins, 1989). The difference between the two estimator is that Watkin's Q $(\lambda)$ terminates the calculation of $\lambda$-return whenever an exploratory action is taken. In our experiments, we adopt the configuration that uses Peng's Q $(\lambda)$, since it has been demonstrated in (Daley & Amato, 2019) that Peng's Q $(\lambda)$ is superior to Watkin's Q $(\lambda)$.

### A1.2    SALIENCY MAP GENERATION METHODOLOGY OF THE ATTENTION MAPS FOR MB-DQN

In order to perform the qualitative analysis for comparing MB-DQN with the bootstrapped DQN in Seciton 4.1, we utilize the methodology proposed in  (Greydanus et al., 2018) to construct perturbation-based saliency maps to highlight what an agent actually perceives and focuses on from its observation. This information offers a way to understand the learning procedure of the agent,

and enables us to validate that bootstrapped heads with different backup lengths may concentrate on different subjects in an agent's observation. Given an image $I$, the method first generates a the perturbed image $I'$ defined as:

$$I'(i, j) = I * (1 - M(i, j)) + A(I, \sigma_A) * M(i, j), \tag{A2}$$

where $M$ is the mask centered at the coordinate $(i, j)$ of $I$, and $A$ is the Gaussian blur operator with a standard deviation $\sigma_A$. The aim of the perturbation process is to add uncertainty, in order to cause the agent become more uncertain about the area around $(i, j)$. The saliency map $S$ is then defined as the L2-norm difference between the estimated values of $I$ before and after each perturbation process:

$$S(i, j) = ||V(I) - V(I'(i, j))||^2, \tag{A3}$$

The L2-norm difference serves as a measure to reflect the extent of attention of the agent around region $(i, j)$. The saliency map $S$ can be added to one of the three color channels of the original image $I$ to visualize the attention map of the agent. In order to construct saliency maps for MB-DQN, the methodology discussed above is extended to a bootstrapped version, in which all the bootstrapped heads that participate in the final decision of a taken action (denoted as $a_{voting}$) are taken into consideration (i.e., the bootstrapped heads contributing to the highest votes in the majority voting procedure). The saliency measure of each bootstrapped head $k \in K$ (where $K$ is the total number of heads) is then defined as $S_k$, and the total saliency map $S_{bootstrapped}$ for MB-DQN is thus defined as:

$$S_{bootstrapped} = \sum_{k=1}^{K} S_k x_k, \text{where } x_k = \begin{cases} 1, & \text{if } a_k = a_{voting} \\ 0, & \text{otherwise} \end{cases} \tag{A4}$$

## A2 ADDITIONAL DETAILS OF THE EXPERIMENTAL SETUP

In this section, we provide additional training details of our experiments. The agents are evaluated based on the average scores of ten test episodes every 250k timesteps. During the evaluation phase, both the proposed MB-DQN and bootstrapped DQN use a majority vote policy to decide the action to be taken. In other words, every bootstrapped head predicts its own action for each input state, while the action taken by the agent is determined by the highest vote from all of the bootstrapped heads. The experimental results presented in this paper are generated based on three different random seeds, and the evaluation curves are drawn with 68% confidence interval (i.e., one standard deviation) as the shaded areas. The detailed settings of the hyper-parameters in our experiments are listed in Table A1.

### A2.1 THE HYPERPARAMETERS FOR TRAINING MB-DQN

Table A1 summarizes the hyper-parameters of MB-DQN, Bootstrapped DQN (Osband et al., 2016), and DQN($\lambda$) (Daley & Amato, 2019), including the training parameters and their configurations, followed by the input sizes for the agents.

### A2.2 NETWORK STRUCTURE

In this section, we explain the network structure used in our experiments. The shared convolutional neural network depicted in Fig. 2 of the main manuscript consists of three convolutional layers with 32, 64, and 64 filters, which is the same as the configurations used in DQN (Mnih et al., 2015), Bootstrapped DQN (Osband et al., 2016), and DQN ($\lambda$) (Daley & Amato, 2019). The output of the last convolutional layer is then fed into $K$ distinct heads, where each head is implemented as a fully-connected (FC) layer with 512 filters, followed by another FC layer to predict Q-value for each action. The default activation function is set to ReLU.

## A3 ADDITIONAL EXPERIMENTAL RESULTS

In this section, we provide and discuss additional experimental results evaluated on *Atari 2600* games in Section A3.1, as well as additional sets of ablation analyses of the backup lengths in Section A3.3.

Table A1: The detailed settings of the hyper-parameters using in our experiments.

| Hyperparameter | Value |
|---|---|
| **Mixture of Step Returns in Bootstrapped DQN (MB-DQN)** | |
| Learning rate of the agent | 2.5e−4 |
| Discount factor ($\gamma$) | 0.99 |
| Optimization for RL agent | RMSProp |
| Minibatch size | 32 |
| Target update period | 1 update per 10000 timesteps |
| Training timesteps | 10M |
| Replay buffer size | 1M |
| Number of the bootstrapped heads | 10 |
| Head return $n_k$ (Algorithm 1) | 1 step * 5 heads, |
| 3 steps * 5 heads | |
| **Bootstrapped DQN** (Osband et al., 2016) | |
| Learning rate of the agent | 2.5e−4 |
| Discount factor ($\gamma$) | 0.99 |
| Optimization for RL agent | RMSProp |
| Minibatch size | 32 |
| Target update period | 1 update per 10000 timesteps |
| Training timesteps | 10M |
| Replay buffer size | 1M |
| Number of the bootstrapped heads | 10 |
| Head return $n_k$ | 1 step * 10 heads (default), |
| varies for different experiments | |
| **DQN($\lambda$)** (Daley & Amato, 2019) | |
| Learning rate of the agent | 1e−4 |
| Discount ($\gamma$) | 0.99 |
| Optimization for RL agent | Adam |
| Minibatch size | 32 |
| Replay buffer size | 1M |
| Cache size | 80000 |
| Block size | 100 |
| Return estimator | Peng's Q($\lambda$) (Peng & Williams, 1994) |
| $\lambda$ | 0.75 |
| **Input Size for the RL agents** | |
| Input size in *Dense Maze* | current position (x,y), where $0 \leq x, y < 40$ |
| Input size in *Atari* | 84*84*4 |

### A3.1 Additional Evaluation Results on Atari 2600 Games

Following the evaluation results presented in Section 4.1 of the main manuscript, in this section, we provide the the evaluation results of the Bootstrapped DQN baselines and MB-DQN on 33 *Atari* games in Table A3, where these methods are similarly denoted as *All-1-Step (Baseline)*, *All-3-Step (Baseline)*, and *Mixed-1-3-Step (MB-DQN)*, respectively. The settings of the Bootstrapped DQN baselines and MB-DQN are the same as those in Section 4.1 of the main manuscript. The scores highlighted in bold represent the best performances among these three methods. It can be observed that MB-DQN achieves the best score in 19 out of the 33 games, while *All-3-Step (Baseline)* outperforms the other two in 11 out the 33 games. In contrast, *All-1-Step (Baseline)* only wins in the remaining three games.

### A3.2 Additional Examples of the Attention Maps from the MB-DQN Agents

Following the results presented in Section 4.1 of the main manuscript, we further visualize an additional set of attention maps generated by MB-DQN in six *Atari* games, including *Frostbite*, *Alien*, *BeamRider*, *Enduro*, *Riverraid*, and *Amidar*, and present their results in Fig. A1 (a)-(f), respectively.

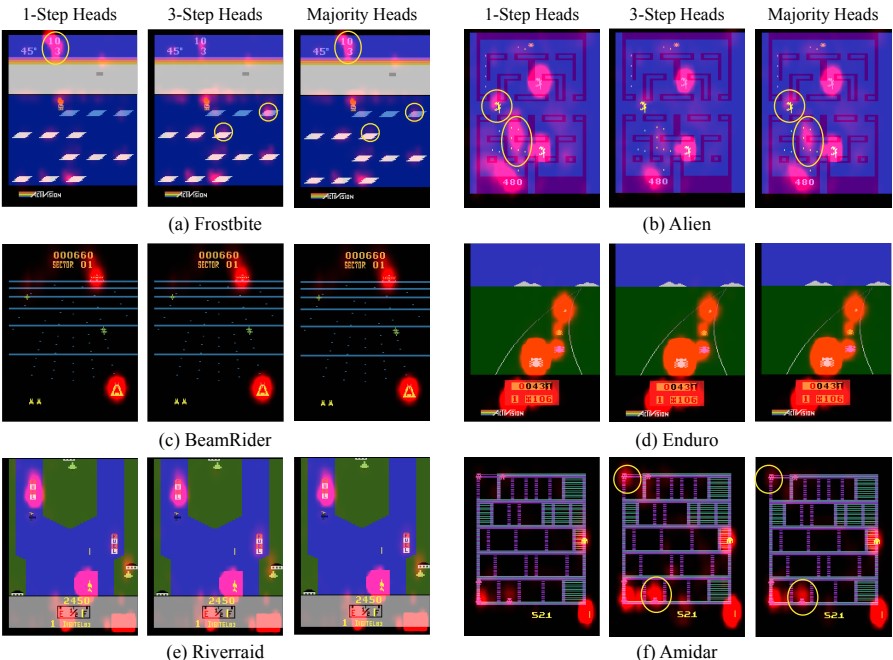

Figure A1: Visualization of the agents' attention areas (rendered in red) for six *Atari* games.

From Fig. A1, it is observed that in three out of the six games (i.e., *Frostbite*, *Alien*, and *Amidar*), the attentions of *1-Step Heads* and *3-Step Heads* focus on different regions. On the other hand, in the remaining three games (i.e., *BeamRider*, *Enduro*, and *Riverraid*), the attention areas of *1-Step Heads* and *3-Step Heads* are quite similar. Please note that even though the attention areas of them are similar in some games, the agents might still exhibit different behaviors that result in different evaluation scores, as shown in Table. A3. This is due to the fact that the visualization approach proposed in (Greydanus et al., 2018) is only a way to interpret the behaviors of the agents, and are thus unable to reflect the full behaviors of them. In the following paragraphs, we provide further discussions of the three games in which the attention regions of the *1-Step Heads* and the *3-Step Heads* are different in MB-DQN.

***Frostbite.*** The *1-Step Heads* maintain focusing on the scoreboard, which might be a sign of overfitting. This might be the reason that causes the poor performance of *1-Step Heads* in Table. A3.

***Alien.*** In order to achieve a high score in *Alien*, the agent has to learn to dodge the attacks from the aliens, survive, and then destroy the alien eggs laid in the hallways. It seems that *1-Step Heads* concentrate on both the eggs and the monster with a similar extent of attention (i.e., the red colors of the attention regions are highlighted with similar magnitudes), which might cause the agent to be overly greedy in destroying the eggs, leading to its death and undesirable performance in Table. A3.

***Amidar.*** From Fig. A1 (f), it is observed that the *3-Step Heads* seem to pay more attention on the monsters than the *1-Step Head*. *3-Step Head* can easily walk around the grid to get higher rewards without attacking by monsters. On the contrary, the *1-Step Heads* merely concentrate on the character.

For these three games, the attention areas of the *Majority Heads* cover the areas from both the *1-Step Heads* and the *3-Step Heads*, allowing MB-DQN to outperform the two baselines *All-1-Step (Baseline)* and *All-3-Step (Baseline)* in Table. A3. These examples again validate that MB-DQN can leverage the benefits from different backup lengths as discussed in Section 4.1 of the main manuscript.

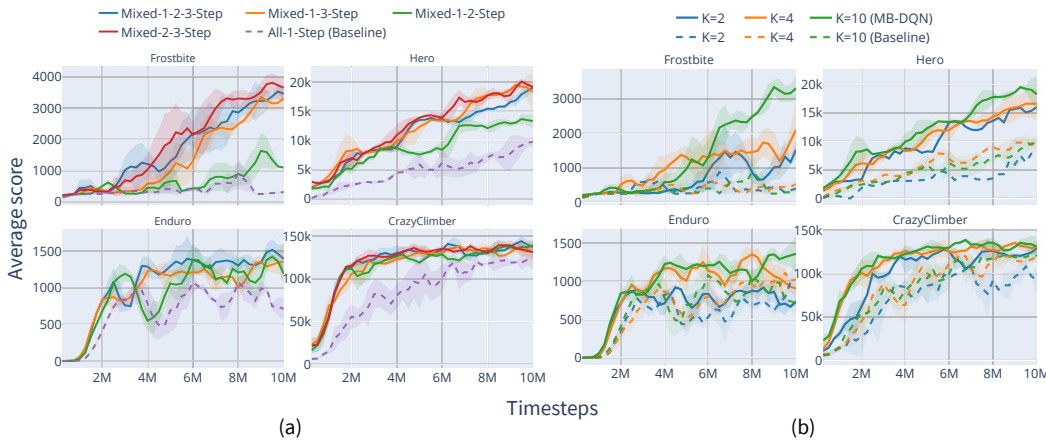

(a)                                                                                    (b)

Figure A2: Curves of four more Atari games for (a) different configurations of step returns for the bootstrapped heads, and (b) different numbers of the bootstrapped heads on the proposed MB-DQN.

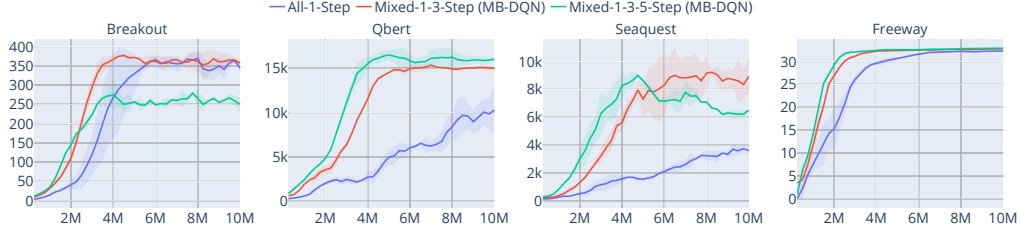

Figure A3: Experimental results of different configurations of step returns in four other *Atari* games.

### A3.3 Additional Results for Ablation Analyses

In this section, we present more results for Section 4.4 of the main manuscript, and consider different upper bounds of step returns for the bootstrapped heads in MB-DQN. In Fig. A2, we depict the learning curves of four more *Atari* games, including *Frostbite*, *Enduro*, *Hero*, and *CrazyClimber*. Fig. A2 (a) shows that the agents trained with different mixtures of step returns perform better than the *All-1-Step* baseline. Fig. A2 (b) illustrates the advantage of the mixture usage of multi-step returns in bootstrapped heads. These additional results lead to similar insights, as discussed Section 4.4.

We further inspect in detail three different configurations of backup lengths for the bootstrapped heads in MB-DQN, including *All-1-Step (Baseline)*, *Mixed-1-3-Step*, and *Mixed-1-3-5-Step*. The learning curves of these three configurations are presented in Fig. A3. It can be observed that the agents trained with a larger upper bound of backup lengths (i.e., *Mixed-1-3-5-Step*) learn faster in three out of the four games, including *Qbert*, *Seaquest*, and *Freeway*. An observation from the results is that the agents suffer from performance drops in *Seaquest* and *Breakout* for the *Mixed-1-3-5-Step* setting. This is caused by the implication revealed in Fig. 3 and Section 4.1 of the main manuscript that longer backup lengths might not necessarily bring beneficial impact on the learning process of the value function. The observations suggest that the optimal configuration of the bootstrapped heads and the upper bound of the backup length are still a challenging issue to be investigated in the future.

## A4 Computing Infrastructure

In this section, we provide the configuration of our computing infrastructure in Table. A2 for reference.

Table A2: Specification of our computing infrastructure.

| Component | Customized Machine |
|---|---|
| Processor | 32 cores / 64 threads (3.0GHz, up to 4.2GHz) |
| Hard Disk Drive | 6TB SATA3 7200rpm |
| Solid-State Disk | 1TB PCIe Gen 3 NVMe |
| Graphics Card | NVIDIA GeForce® RTX 2080Ti (two cards per instance) |
| Memory | 16GB DDR4 2400MHz (128GB in total) |

## A5 REPRODUCIBILITY

We implemented the proposed MB-DQN based on the RLTF framework (Nikolov, 2018), which is a research framework that provides high-quality implementations of common RL algorithms based on the TensorFlow deveplopment platform. We modified the source codes of RLTF and added an additional option *MB-DQN* into its DQN family. All the conducted experiments presented in our paper are re-producible with easy-following instructions. For more details about our source codes, please refer to the anonymous github repository at the following link: https://github.com/Anonymous-Source-Code/MB-DQN.

Table A3: Comparison of the evaluation results of MB-DQN and the baselines in 33 *Atari* games.

| | All-1-Step (Baseline) | All-3-Step (Baseline) | Mixed-1-3-Step (MB-DQN) |
|---|---|---|---|
| Alien | 1411.7 | 2633.7 | **3486.7** |
| Amidar | 344.6 | 544.1 | **639.1** |
| Asterix | 6190 | 6431.7 | **7200.0** |
| BankHeist | 812.7 | 977.3 | **1215.7** |
| BeamRider | 9832.3 | **13576.5** | 13440.6 |
| Bowling | 36.8 | **47.2** | 30.8 |
| Boxing | 81.8 | 90.8 | **91.1** |
| Breakout | 315.4 | 87.5 | **368.6** |
| CrazyClimber | 121587 | 49313.3 | **133446.7** |
| DemonAttack | 14214.8 | 9532 | **14292.0** |
| Enduro | 1077 | 526.7 | **1365.1** |
| Freeway | 32.2 | 32.8 | **32.9** |
| Frostbite | 300 | 2765.7 | **3417.0** |
| Gopher | **8664.7** | 3640 | 4376.7 |
| Gravitar | 10 | **200.0** | 58.3 |
| Hero | 10039.2 | 13033.2 | **18318.0** |
| Kangaroo | 2646.7 | **14993.3** | 12520 |
| Krull | 6874.6 | 7861.6 | **9261.4** |
| KungFuMaster | 15026.7 | 10583.3 | **27930.0** |
| MontezumaRevenge | 0 | **3.3** | 0 |
| MsPacman | 2264 | **2508.0** | 2469 |
| Pong | 20.7 | **20.9** | **20.9** |
| PrivateEye | 27.9 | **140.0** | -50.8 |
| Qbert | 11065 | 13165.8 | **15083.8** |
| Riverraid | 11651 | 13102.3 | **14154.7** |
| RoadRunner | **49053.3** | 46740 | 38740 |
| Seaquest | 2584 | 4407.3 | **8842.0** |
| Skiing | -23763.8 | **-19122.9** | -23199.2 |
| Solaris | **807.0** | 327.3 | -50.8 |
| SpaceInvaders | 1505.8 | 1279 | **1615.0** |
| Venture | 0 | **266.7** | 66.7 |
| WizardOfWor | 983.3 | **4600.0** | 2010 |
| Zaxxon | 106.7 | **6826.7** | 2963.3 |

