# OpenReview forum: "Mixture of Step Returns in Bootstrapped DQN"
_ICLR.cc/2021/Conference — Reject_

### Official Review · AnonReviewer1 · 2020-10-25
**Interesting combination of two ideas, but some concerns to be discussed..**

**Rating:** 5
**Confidence:** 5

**Review:**

###################################

Summary:

This paper combines the idea of multi-step returns for value-based reinforcement learning with Bootstrapped DQN. Updating value functions with different step returns has potential benefits, and was elegantly integrated with the Bootstrapped DQN structure, where different heads have different backup lengths. This work provides empirical results on Atari game domains, and further provides ablation analysis.



###################################

Pros:

1. This paper presents a simple, but meaningful approach to combine multi-step returns in value-function based RL and Bootstrapped DQN. The strength of having different backup length in value-based RL is that multi-step returns have lower bias (despite the high variance) and sensitivity to future rewards. The power of Bootstrapped DQN comes from its exploration capabilities. Bootstrapped DQN’s multiple heads are a natural way of integrating different lengths of returns. I like the originality of this idea presented by this paper.

2. This paper provides good empirical results in Atari games, showing that mixed 1-3 step MB-DQN outperforms fixed-step baselines.

3. This paper also provides ablation studies, adding more comprehensive analysis of this algorithm. Separating data generation agent and learning-only agent is valid; MB-DQN is compared with DQN-lambda and DQN-ensemble and shows better performance; and more experimental results with different numbers of heads and different configurations of return lengths are compared.


###################################

Cons / Questions

1. The choice of (step length) n_k is limited: this paper presented main empirical results using five heads of 1-step and five heads of 3-step, and also compared with mixed 1-2-3, 1-2, 1-3, 2-3 steps. In the supplementary materials, there is a result on 1-3-5 step configurations. What about results on larger n_k steps? Could you add some experiments on 10 or 20 steps or at least add insights on how it would perform? My guess is that including very large n_k to the mixture would lead to high variances and possibly degrade the performance.

2. The original strength of Bootstrapped-DQN comes from the fact that it is better for exploration. (Multiple heads lead to explorations.) However, I don’t see comprehensive analysis in the perspective of explorations. To be specific, as authors mentioned in section 3.1., shorter-step return is better at exploring wider areas in the state spaces than longer-step returns, although longer-step returns might be faster at convergence. In terms of exploration perspective, is there an evidence that mixture of shorter & longer n_k’s is better than single fixed 1-step returns?

###################################

Reasons for Score:

I gave the score of 5 (marginally below the threshold). But if the concerns that were raised above are well addressed, I am happy to change my score to a higher score after the rebuttal period. Overall, I appreciate the novelty of this paper, and I think the idea of combining two powerful methods in value-based RL: multi-step returns and bootstrapped DQN is meaningful.

---

> ### Author Response · Authors · 2020-11-21
> **Response to Reviewer #1**
>
> We sincerely thank the reviewer for spending time and providing valuable feedback. We would like to address the concerns from the reviewer by providing our responses as well as our additional experimental results.
>
> Q1. The choice of (step length) n_k is limited: this paper presented main empirical results using five heads of 1-step and five heads of 3-step, and also compared with mixed 1-2-3, 1-2, 1-3, 2-3 steps. In the supplementary materials, there is a result on 1-3-5 step configurations. What about results on larger n_k steps? ***Could you add some experiments on 10 or 20 steps or at least add insights on how it would perform?*** My guess is that including very large n_k to the mixture would lead to high variances and possibly degrade the performance.
>
> A1. We appreciate the constructive suggestion.  ***We agree with the reviewer and have performed experiments using a new configuration Mixed-1-5-10-Step***, which contains three heads of 1-step, three heads of 5-step, and four heads of 10-steps.  The results of this configuration are compared with ***Mixed-1-3-Step***, ***Mixed-1-3-5-Step***, and ***All-1-Step (Baseline)*** in the following figure.
>
> > ***The additional experimental results with a new configuration Mixed-1-5-10-Step*** in the following link: https://imgur.com/a/kiGuURK
>
> It can be observed that this configuration does lead to degraded performance in Breakout and Seaquest, as mentioned by the reviewer.  However, we also found that for Freeway and Qbert, this configuration delivers the best performance than ***Mixed-1-3-Step*** and ***All-1-Step (Baseline)***.
>
> This observation reveals that mixing the backup lengths is beneficial, however, the range of backup lengths should not be overly large or too small, and may depend on the environment.
>
> ---
>
> Q2. The original strength of Bootstrapped-DQN comes from the fact that it is better for exploration. (Multiple heads lead to explorations.) However, I don’t see comprehensive analysis in the perspective of explorations. To be specific, as authors mentioned in section 3.1., shorter-step return is better at exploring wider areas in the state spaces than longer-step returns, although longer-step returns might be faster at convergence. In terms of exploration perspective, ***is there an evidence that mixture of shorter & longer nk’s is better than single fixed 1-step returns?***
>
> A2. We appreciate the reviewer for raising this question. We are glad to leverage the grid-world setting presented in Section 3.1 of our manuscript to compare the exploration behaviors in terms of the stated visited for the following four configurations: (a) All-1-Step, (b) Mixed-1-5-agent (corresponding to MB-DQN), (c) All-5-Steps, and (d) DQN ($\lambda$).
>
> > ***We provide the additional experimental results to compare the exploration behaviors*** in the following link: https://imgur.com/a/FqnndK0
>
> It can be observed that the states visitation of the ***Mixed-1-5 agent*** covers those visited by the ***All-1-Step*** and ***All-5-Step*** agents.  In addition, the experiments on the quality of data samples presented in Section 4.2 also validate the exploration benefits of MB-DQN indirectly from a different perspective. The above experiments can therefore serve as an interpretation for the impacts of MB-DQN on the exploration behavior, which is more diverse and is benefited from different bootstrapped lengths.  The relatively higher performance in Fig. 5 (a) in the manuscript shows that the data samples generated by MB-DQN are better in quality than those generated by the other configurations.
>
> In addition to the results in Section 4.2, it is observed that the attention areas of the heads with different backup lengths presented in Fig. 4 of the original manuscript focus on different regions. The difference in the behaviors might be caused by the difference in exploration during the training phase.
>
> We hope that the above explanations have adequately addressed the reviewer's concerns.

---

> > ### Comment · AnonReviewer1 · 2020-11-24
> > **Reply to the response**
> >
> > Thank you for the detailed answers to my questions. My major concerns have been fully addressed by the authors. It's interesting to see that, in some domains, if large n_k is used, then the performance might degrade. It's also notable that the optimal range of mixture of step lengths might differ depending on domain. Also, thank you for the additional experiments on the exploration behaviors: it definitely seems that the mixed 1-5 agent covers wider regions than single-fixed-step length agents (either 1-step or 5-step agent).

---

> > > ### Author Response · Authors · 2020-11-24
> > > **Response to Reviewer #1**
> > >
> > > We sincerely appreciate the prompt reply from the reviewer and would like to thank the reviewer again for spending time and efforts reviewing this paper, as well as providing constructive comments for this work.

---

### Official Review · AnonReviewer4 · 2020-10-27
**The paper introduces different step return targets to bring more heterogeneity to the bootstrapped DQN, but the improvement is not significant enough.**

**Rating:** 4
**Confidence:** 4

**Review:**

This paper utilizes different step return targets for different heads in bootstrapped DQN, such that a more heterogeneous posterior estimation of policy value may be obtained. However, my main concern is the novelty. The step size can be viewed as a tunable hyperparameter, and the posterior computation is mainly credited to the usage of different randomized DQNs. Similarly, other tunable hyperparameters may also be diversified to introduce more heterogeneity such as learning rates, etc.

Re- simulations, I appreciate the authors' efforts in conducting different experiments to understand their method, but I do have a few comments/thoughts as below:
- what is the performance of all-2-step bootstrapped DQN? It seems more natural to me to include the comparison with this baseline, since the performance of mixed-1-3 may be more close to all-2.
- I'm not sure if Figure 4 does do you a favor, since again we can always tune the hyperparameter of step size, and it does not need to be the same for different environments. This paper's method seems to not peak either of them.
- the reason behind the good performance of MB-DQN explained in section 4.2 is not obvious to me. I'm not sure about the point of comparing an agent trained with its own data versus one trained with other data, since we are just learning the policy value but not the policy itself, and there is no issue like off-policy correction, etc. Actually such techniques of data-sharing have been used a lot if we have multiple agents to collect data.
- in section 4.4, when comparing different setups of MB-DQN, it would be fairer to include more baselines used in part of the MB-DQN such as all-2, all-3.

---

> ### Author Response · Authors · 2020-11-21
> **Response to Reviewer #4 (part 3/3)**
>
> Q4. ***What is the performance of all-2-step bootstrapped DQN?*** It seems more natural to me to include the comparison with this baseline, since the performance of mixed-1-3 may be more close to all-2.
>
> > ***We provide the additional experimental results with All-2-Step setting*** in the following link: https://imgur.com/a/KkMpoDm
>
>
> A4. Thanks for raising this question.  In order to address the reviewer's concern, we provide the following set of figures, which consists of two sets of experiments: ***(a) All-1-Step / All-2-Step / All-3-Step / Mixed-1-3-Step*** and ***(b) All-3-Step / Mixed-1-3-5-Step***.  The settings are based on the same architecture containing ten bootstrapped heads.
>
> In the first set of experiments, the results are similar to the reviewer's guess, in which the All-2-Step agent performs closely to the Mixed-1-3-Step case except for the Sequest environment. On the other hand, in the second set of experiments, it is observed that the All-3-Step agent does not outperform the Mixed-1-3-5-Step agent in three out of the four games.  These additional sets of experiments reveal two insights.  First, averaging the step sizes does not necessarily result in the same performance as the heterogeneous design. Second, selecting an appropriate (or optimal) backup length might be not a straightforward task, as it may change from environment to environment.
>
> A key advantage of the heterogeneous design is that ***only a range of backup lengths (e.g., 1-3, 1-3-5, or 1-5-10) has to be defined, rather than a specific value that might necessitate a hyperparameter search phase.***  Moreover, the experimental results discussed in Section 4.2 also show that the data samples collected by the heterogeneous design is superior to those collected by the agent based on a single return target.  The results in Fig. 4.3 further demonstrates that the single target design based on DQN-lambda is not as satisfactory as MB-DQN.  As a result, we believe that the impacts of a heterogeneous design are different from a single return target.
>
> ---
>
> Q5. In section 4.4, when comparing different setups of MB-DQN, it would be fairer to include more baselines used in part of the MB-DQN such as all-2, all-3.
>
> A5. We appreciate the suggestion from the reviewer. We have the results performed on various configurations (such as the All-2 case depicted above, the All-3 case in the original manuscript, as well as the results presented in our responses), and would be glad to incorporate them all in the final version

---

> ### Author Response · Authors · 2020-11-21
> **Response to Reviewer #4 (part 2/3)**
>
> Q2. The reason behind the good performance of MB-DQN explained in section 4.2 is not obvious to me. ***I'm not sure about the point of comparing an agent trained with its own data versus one trained with other data***, since we are just learning the policy value but not the policy itself, and there is no issue like off-policy correction, etc. Actually, such techniques of data-sharing have been used a lot if we have multiple agents to collect data.
>
> A2. We are afraid that there might be some ***misunderstanding***. ***We are actually not "comparing an agent trained with its own data versus one trained with other data".***
>
> In fact, what we are comparing are as follows:
>
> (a)
>
> Data generatored from ***Mixed-1-3 (MB-DQN)*** -> Evaluated by ***All-3-Step***
>
> Data generatored from ***All-1-Step*** -> Evaluated by ***All-3-Step***
>
> (b)
>
> Data generatored from ***Mixed-1-3 (MB-DQN)*** -> Evaluated by ***All-1-Step***
>
> Data generatored from ***All-3-Step*** -> Evaluated by ***All-1-Step***
>
> In these two experiments, the data collection agent and the learning agent are different. One agent (i.e., the data generation agent) is responsible for generating state-action pairs for an experience replay buffer while updating its Q-value network with the data contained in it. The other agent (i.e., the evaluation agent) only updates its Q-value network by the existing data samples contained in the replay buffer, without contributing data to it.
>
> The significance of this set of experiments lies in the validation of the quality of the data samples generated by MB-DQN. Taking (a) for instance, if the data samples of MB-DQN is better than those generated by the All-3-Step agent, the performance of the All-1-Step agent trained by the data samples from the former would also be better than the latter.  The set of experiments in (b) validates the quality of data samples using the All-3-Step agent for evaluation.  From Fig. 5 (a), it is observed that both experiments (a) and (b) indicate that the data samples generated by MB-DQN are beneficial.  As a result, it validates the assumption made in Sections 1 and 2.  As the quality of data samples plays an essential role in the learning process of the agent, the experiments explain why the MB-DQN agents are able to achieve superior performance than the baselines.
>
> We would like to bring to the reviewer's kind attention that the quality of data samples is not only beneficial to the learning of the value functions but also contributes to the learning of the agent's policy. The learning processes of the policy and the value function are actually two sides to the same coin.
>
> ---
>
> Q3. I'm not sure if Figure 4 does do you a favor, since again we can always tune the hyperparameter of step size, and it does not need to be the same for different environments. ***This paper's method seems to not peak either of them.***
>
> A3. We are afraid that there might be some ***misunderstanding***.
>
> The core theme of this paper is the investigation of the impacts of the backup lengths on the exploration behaviors, the performance, as well as the quality of the collected data samples, rather than to peak either of them.
>
> We would also like to bring to the reviewer's kind notice that the performance of MB-DQN is actually superior to the All-1-step and the All-3-step baselines in these two games.  ***Fig. 4 is included in the paper to explain the rationale behind the superior performance from MB-DQN.***  From the perspective of exploration, it is favorable to have more data samples collected from different behavioral policies. ***The inclusion of Fig. 4 provides an explanation and serves as evidence that the heads with different backup lengths may lead the agent to behave differently.***
>
> We sincerely hope that the above explanation helps to address the concern.

---

> ### Author Response · Authors · 2020-11-21
> **Response to Reviewer #4 (part 1/3)**
>
> We sincerely thank the reviewer for spending time and providing valuable feedback. We would like to address the concerns from the reviewer by providing our responses as well as our additional experimental results.
>
> Q1. my main concern is the novelty. The step size can be viewed as a tunable hyperparameter, and the posterior computation is mainly credited to the usage of different randomized DQNs. Similarly, other tunable hyperparameters may also be diversified to introduce more heterogeneity such as learning rates, etc.
>
> A1. We would like thank the reviewer for raising this question, and would like to clarify the difference in terms of two different perspectives.  First, we agree with the reviewer that adopting different hyperparameters such as different learning rates may also introduce heterogeneity and cause different exploration behaviors.  However, ***please kindly note that changing the learning rates (or other hyperparameters) does not change the targets of the Q-functions.***  In other words, all heads would approximate the same target.  The difference among the heads thus becomes the magnitude of the gradients used for updating them.  Although the heterogeneity in other hyperparameters may also cause different hebehaviors among the bootstrapped heads, this direction seems misaligned with the ***core theme of this paper - discussing the impacts of the return targets on the exploration behaviors, the performance, as well as the quality of the collected data samples.***
>
> Second, discussing the heterogeneity in other hyperparameters would prevent us from comparing with the conventional TD ($\lambda$) approach and the more recent DQN ($\lambda$) method fairly. To systematically identify which is the most effective integration method of combining different return targets, it is essential to keep the other hyperparameters the same, and only allowing the return target of the Q function to be altered.  We agree that changing the other hyperparameters would also be an interesting research direction, nevertheless, it might not be suitable in the context of comparing the return function design methods.
>
> In order to justify our proposed concept, a comprehensive set of experiments are performed for different lengths of step returns as well as various fasions of combinations.  In addition, a set of gridworld experiments as well as attention areas are incorporated in the comparison of MB-DQN and the baselines.  From this perspective, we believe that the merits from this paper are the in-depth investigations of the return target design, which have not been properly discussed in the literature before.
>
> We sincerely hope that the above explanation has properly addressed the reviewer's concern.

---

### Official Review · AnonReviewer2 · 2020-10-28

**Rating:** 4
**Confidence:** 4

**Review:**

### Summary of Contributions

The paper proposes the Mixture Bootstrapped DQN (MB-DQN) algorithm, an extension of bootstrapped DQN where each outputted value estimate uses a different multi-step TD target. They provide a motivating example suggesting that with shorter backups, the slower convergence results in greater exploration. They empirically show that in a variety of cases, simultaneously learning action value estimates with varying multi-step TD targets can lead to improvements over existing methods.

### Review

The proposed algorithm appears novel, and the results are interesting. However, I have the following concerns:

1) A high level description was provided on how actions were selected during evaluation, but none was provided as to how they were selected during training (i.e., the behavior policy). This information is important given the use of Peng's Q(λ) targets (*uncorrected* returns), as the update targets are now a mixture of on-policy sampled rewards, and off-policy bootstrapped values. Could the authors elaborate on the behavior policy used while training, and comment on its compatibility with the use of uncorrected multi-step returns?

2) The motivating example compares learning with a 1-step TD target with that of a 5-step TD target. The paper suggests that due to the larger bias in a 1-step TD target (and consequently, slower convergence), the 1-step learner ends up exploring more. Especially in the 1-step case (where Peng's Q(λ) does not need corrections), the algorithm learns off-policy, making it seem like whether or not it explores more an artifact of a design choice in the behavior policy, and not something inherent about 1-step TD targets. Also of note, when looking at the heatmaps after 100,000 time steps, the 1-step learner has a very salient line taking a suboptimal path of going straight up for a bit, and then heading in a straight line toward the goal. Such a high frequency of this suboptimal path seems suggestive of the method actually *not* exploring enough. As such, I'm not sure the conclusions drawn from the example are convincing, and would appreciate if the authors could comment on the above or clarify any misunderstanding.

3) In addition to the motivating maze example, the paper repeatedly emphasizes that MB-DQN provides heterogeneity in the target values. Can the authors clarify what this precisely means, and elaborate on why this is a desirable property?

4) It appears a fixed learning rate was used for each algorithm. Due to the bias-variance tradeoff of multi-step TD methods, as well as the varying numbers of heads in the ablation studies, it's not clear whether a fair comparison was made between each algorithm instance. Could the authors comment on how hyper-parameters were chosen in the empirical evaluation?

5) Only 3 random seeds were used in their empirical evaluation, and the paper presents one standard deviation as a 68% confidence interval. The use of 3 seems insufficient for central limit theorem arguments, but the mention of 68% suggests an assumption that the results are normally distributed. Can the authors comment on the statistical significance of their results, and whether the number of runs are sufficient for the claims being made?

Based on the above, I'm recommending rejection at this time. I'm willing to raise my score should my concerns be addressed.

----- Post Discussion -----

I appreciate the clarifications made in the discussion regarding additional experimental details and whether the methods were fairly compared. However, given that much of the claims rely heavily on the empirical evaluation, I think further experiments with more rigorous statistical analysis is necessary. Even with the correctly plotted standard errors, the results still largely do not appear significant, suggesting that 5 seeds is just not enough. There are good recommendations by Henderson et al. (2017) and Colas et al. (2018) for the empirical evaluation, and I think it would additionally strengthen the paper to formalize the notion of heterogeneity (e.g., be able to approximately measure it, and convincingly argue that this is what's underlying any differences in performance).

---

> ### Author Response · Authors · 2020-11-21
> **Response to Reviewer #2 (part 2/2)**
>
> Q4. In addition to the motivating maze example, the paper repeatedly emphasizes that MB-DQN provides heterogeneity in the target values. ***Can the authors clarify what this precisely means, and elaborate on why this is a desirable property?***
>
> A4. We are glad to clarify the meaning of heterogeneity. As described in Section 3, the original Bootstrapped DQN employs an architecture in which the backup length of different bootstrapped heads is the same.  In contrast, the core theme of MB-DQN is the adoption of different backup lengths for different bootstrapped heads.  This is where "heterogeneity" comes from.  It means that the bootstrapped heads are trained in different manners.
>
> As illustrated in Fig. 1 and explained in Section 3.1, DQN agents trained with different backup lengths may have different behavior policies.  The examples in the grid-world maze environment illustrate that one step learners tend to explore widely in the environment, while the five-step learner quickly converges to its optimal policy.  Therefore, the behaviors as well as the experiences of the agents are different. MB-DQN enables the agent to benefit from the properties of different lengths of step returns. We have additionally provided the experimental results on the 2D maze environment to show the behavior of the agent trained with a mixture of multi-step returns.
>
> > ***We provide the additional experimental results*** in the following link: https://imgur.com/a/FqnndK0
>
> A visualization of the attention areas depicted in Fig. 4 also validates the above observation from another perspective. As explained in Section 4.1, the focused areas of the bootstrapped heads with different backup lengths fall in different regions. This explains why the behaviors of the agents trained by different backup lengths are different.
>
> The final perspective is from the collected data.  In the experimental results presented in Section 4.2, it is observed that the data samples collected by a fixed backup length (i.e., the way used by the original Bootstrapped DQN) are not superior to those collected by MB-DQN.  In addition, the results from the grid-world maze environment also illustrate that the behaviors caused by different step return targets are different.
>
> Based on the above three perspectives, the experimental evidence and the observations all suggest that heterogeneity in backup lengths is a potentially promising and desirable property for Bootstrapped DQN based architectures.
>
> We sincerely hope that the above explanation can address the question from the reviewer.  Please kindly let us know if there is any other unclear point.
>
> ---
> Q5. It appears a fixed learning rate was used for each algorithm. Due to the bias-variance tradeoff of multi-step TD methods, as well as the varying numbers of heads in the ablation studies, it's not clear whether a fair comparison was made between each algorithm instance. ***Could the authors comment on how hyper-parameters were chosen in the empirical evaluation?***
>
> A5. Thanks for raising the question.  We would like to clarify our rationale base on two perspectives.  First, in order to compare with Bootstrapped DQN in a fair manner, all of the settings in our experiments are chosen to be the same as those described in the original paper [1].  We agree with the reviewer that changing the hyper-parameters such as the learning rate may be beneficial to bootstrapped heads with longer backup lengths.  However, fine-tuning the learning rates for bootstrapped heads with different backup lengths will prevent us from identifying whether the performance gain is resulted from the heterogeneity in backup lengths, or is due to the better learning rate.
>
> The other perspective is that in a paper [2], the authors also evaluated a different number of step returns using the same learning rate.  Their motivation is similarly based on a fair comparison purpose.
>
> Due to the above two perspectives, we select to use the same configuration for the bootstrapped heads instead of finetuning them individually.
>
> We hope that the above clarification addresses the reviewer's concern.
>
> [1] I. Osband *et al.* Deep exploration via bootstrapped dqn.
>
> [2] A. Amiranashvili *et al.* Analyzing the role of temporal differencing in deep reinforcement learning.

---

> > ### Comment · AnonReviewer2 · 2020-11-23
> > **Reply**
> >
> > *"We would like to clarify that at the beginning of each episode, the agent picks up a random bootstrapped head as the behavior policy. It then collects experiences using that policy during the same episode, and the collected experiences are store in the replay buffer, which is shared by all of the bootstrapped heads. As a result, the update target is not a mixture of on-policy sampled rewards. Each head has its own return target with a fixed backup length. Neither correction nor modification is applied to the returns collected by the agent, which is the same as the multi-step approach adopted in RainbowDQN [2]."*
> >
> > When following a head as a behavior policy, is it behaving deterministically greedy with respect to it? Many prior works would derive behavior from value estimates soft-greedily (e.g., ε-greedy, Boltzmann over Q) that it's still not clear how specifically the actions are being selected. I believe the observations about more exploratory behavior may be heavily intertwined with this design choice.
> >
> > Further, based on this description, the update target will include a mixture of on-policy sampled rewards. The behavior of one of the heads might not be the behavior according to another head, but the sampled sequence of rewards are shared by every head when used to compute their multi-step returns. While it's not using uncorrected λ-returns (i.e., Peng's Q(λ)), the approach used in RainbowDQN is similarly an uncorrected n-step return.
> >
> > *"We would like to clarify that the trajectory of the agent in an episode is "optimal" as long as it reaches the target and is composed of "moving up" and "moving right" actions."*
> >
> > Thank you for this clarification!
> >
> > *"We have raised the number of random seeds of MB-DQN and the All-3-Step agents to five and modify the confidence interval to 95%. The statistical trends of the updated curves for MB-DQN and the baselines still support the findings of this research."*
> >
> > Can the authors comment on the updated results? I have looked at the album of results with the new 95% confidence intervals, and the statistical trends do not appear significant given the excessive overlap in the intervals.
> >
> > *"This is where "heterogeneity" comes from."*
> >
> > Thank you for the additional description. I was more wondering whether it was something more formal, and perhaps measurable. While it has heads which implicitly represent different behaviors, I think the unclear part is *why* this would lead to better performance, as it's a little vague what is meant by benefitting from the "properties of different lengths of step returns." I feel the claim about additional exploration isn't very convincing due to DQN being generally off-policy, and one could take the supposedly less exploratory learner but provide a more exploratory behavior policy (e.g., ε-greedy with an annealing ε).
> >
> > Have the authors tried running MB-DQN (multiple heads with different backup lengths), but only behaving according to one of the heads (instead of only learning one head of some backup length)? From the differences observed with different fixed backup lengths, it seems possible that the differences may be due to predicting auxiliary tasks. That is, the varying multi-step update targets backpropagating through the same hidden layers may be having some regularization/representation learning effect.
> >
> > *"However, fine-tuning the learning rates for bootstrapped heads with different backup lengths will prevent us from identifying whether the performance gain is resulted from the heterogeneity in backup lengths, or is due to the better learning rate"*
> >
> > Could the authors clarify this? I'm not sure that comparing different learning rates will prevent this. Rather, it seems that fixing it across algorithms will be what prevents identifying this, as it may be fixed at a poor learning rate for another algorithm. Testing different sources of variability in performance should be what helps identify what the gain can be attributed to (especially when they're possibly dependent), in contrast with not testing them. It would generally paint a clearer and more complete picture if results were reported over a range of the parameter (i.e., an ablation study).

---

> > > ### Author Response · Authors · 2020-11-24
> > > **Response to Reviewer #2 (part 3/3)**
> > >
> > > Q4. Could the authors clarify this? I'm not sure that comparing different learning rates will prevent this. Rather, it seems that fixing it across algorithms will be what prevents identifying this, as it may be fixed at a poor learning rate for another algorithm. Testing different sources of variability in performance should be what helps identify what the gain can be attributed to (especially when they're possibly dependent), in contrast with not testing them. It would generally paint a clearer and more complete picture if results were reported over a range of the parameter (i.e., an ablation study).
> > >
> > > A4. We would like to thank the reviewer for the response.  In order to further address the concerns from the reviewer, ***we would like to provide the following analysis performed on the Seaquest environment***.
> > >
> > > > The experimental results of MB-DQN and the baselines on Seaquest for different learning rates are available at the following link: https://imgur.com/a/6Af0QRM
> > >
> > > In this experiment, three different learning rates are evaluated: 1e-3, 2.5e-4, and 1e-4. We compare only ***Mixed-1-3*** and ***All-3*** for different learning rates, since the learning rate of the ***All-1 (i.e. Bootstrapped DQN)*** case has already been well-tuned to 2.5e-4 in the original paper of Bootstrapped DQN. It is observed that the optimal performance of ***Mixed-1-3*** (i.e., ***Mixed-1-3 (2.5e-4)***) outperforms the optimal performance of ***All-3*** (i.e, ***All-3 (2.5e-4)***). It can also be observed that the ***2.5e-4*** learning rate is optimal among these three learning rate settings for both ***Mixed-1-3*** and ***All-3***.
> > >
> > > We hope that the above explanations can adequately address the reviewer's concerns.

---

> > > > ### Comment · AnonReviewer2 · 2020-11-25
> > > > **Reply**
> > > >
> > > > *"During the training phase, MB-DQN uses an ε-greedy policy, in which the value of  is annealed linearly from 1 to 0.01 over the first 1M timesteps"*
> > > >
> > > > Thank you for this clarification! I think such details are important to include in the paper (at least in the appendices).
> > > >
> > > > *"First, if the performance benefits are mainly due to the better representation learned through the multi-step bootstrapped heads, one can expect that the scores corresponding to different heads would be similar, as the heads share the same feature extraction network."*
> > > >
> > > > I don't think it's necessarily the case that each head would have a similar score, due to the different degrees of bias in their update targets (and fixed-points). I think an interesting observation here is how when comparing to the heads of All-3-step, each head of Mixed-1-3-Step seems to do better and have less variability across the heads, which does seem in support of a better or more widely applicable representation being learned. I do like this table and think it better teases apart what is happening than the existing curves in the main text, assuming enough seeds are used to ensure statistical significance.
> > > >
> > > > *"In this experiment, three different learning rates are evaluated: 1e-3, 2.5e-4, and 1e-4"*
> > > >
> > > > Thank you for these results, they help alleviate concerns about whether the parameters provide a fair comparison.
> > > >
> > > > *"We would like to thank the reviewer for raising this question and would like to elaborate on the results in terms of two aspects. First, from the curves presented in the figure below, the trends of the mean scores (plotted in solid lines) of the three methods (i.e., Mixed-1-3-Step (MB-DQN), All-1-Step, and All-3-Step) show that the mean scores corresponding to MB-DQN are able to rise faster and/or higher than those of the other two baselines in most cases, and is comparable to them in Freeway. Second, from the perspective of the confidence intervals (CIs), it is observed that the CIs of the All-3-Step baseline are wider than those of Mixed-1-3-Step (MB-DQN) in several environments. From the above two perspectives, the statistical results indicate that the mixture usage of multi-step returns may offer benefits in mean scores as well as less variance than the baselines with longer backup lengths."*
> > > >
> > > > I think my primary concerns lie here, as the claims in the work are validated empirically. It's not clear that you can conclude that the means are actually rising faster and/or higher due to the size of the confidence intervals. The results suggest that there's about a 95% chance that the true mean lies *somewhere* in this range (i.e., if you were to repeat the experiment many times, 95% of the intervals will trap the true mean). In this case, with the largely overlapping means, there's still a reasonably likely chance that in many of the environments, the 3-step mean could end up higher than the mixed 1-3 step one. To conclude that the differences in the means are beyond random chance, one would need the intervals to not overlap, or use a less conservative significance test where they may overlap a bit. Regarding the intervals being slightly smaller, this is unclear due to the small number of seeds- Henderson et al. (2017) demonstrated that two sets of 5 random seeds can yield seemingly different curves, suggesting that it's not enough to be representative of the data distribution. While the observations in the results look promising, I think it still falls short on the methodological side of the empirical evaluation.

---

> > > > > ### Author Response · Authors · 2020-11-25
> > > > > **Response to Reviewer #2**
> > > > >
> > > > > We would like to sincerely thank the reviewer again for the prompt reply and would like to respond to the question as follows.
> > > > >
> > > > > ---
> > > > >
> > > > > Q1. I think my primary concerns lie here, as the claims in the work are validated empirically. It's not clear that you can conclude that the means are actually rising faster and/or higher due to the size of the confidence intervals. The results suggest that there's about a 95% chance that the true mean lies somewhere in this range (i.e., if you were to repeat the experiment many times, 95% of the intervals will trap the true mean). In this case, with the largely overlapping means, there's still a reasonably likely chance that in many of the environments, the 3-step mean could end up higher than the mixed 1-3 step one. To conclude that the differences in the means are beyond random chance, one would need the intervals to not overlap, or use a less conservative significance test where they may overlap a bit. Regarding the intervals being slightly smaller, this is unclear due to the small number of seeds- Henderson et al. (2017) demonstrated that two sets of 5 random seeds can yield seemingly different curves, suggesting that it's not enough to be representative of the data distribution. While the observations in the results look promising, I think it still falls short on the methodological side of the empirical evaluation.
> > > > >
> > > > > A1.  We would like to thank the reviewer for pointing out this issue, which allows us to have a chance to take a deeper look into our figures.  After a thorough examination. we would like to sincerely apologize that we have used an incorrect function to draw the figures in our previous reply.  In the previous figures, the curves and the shaded areas were drawn for mean +- std * 2, rather than the standard error of the mean (SEM) for confidence intervals.
> > > > >
> > > > > > ***The updated curves drawn with the standard error of the mean (SEM) function for confidence intervals are provided*** in the following link for the reviewer's reference: https://imgur.com/a/sRrvMb5
> > > > >
> > > > > The new updated curves are illustrated in the figures above.  It is observed that the shaded areas become narrower than those of the previous ones.
> > > > >
> > > > > ***"In this case, with the largely overlapping means, there’s still a reasonably likely chance that in many of the environments, the 3-step mean could end up higher than the mixed 1-3 step one."***
> > > > >
> > > > > From the results presented in the updated figures, it is specifically observed that the overlapped regions of ***Mixed-1-3-Step*** and ***All-3-Step*** shrinks.  In Seaquest, there are several overshoot regions for the ***Mixed-1-3-Step*** case when compared with the ***All-3-Step*** case, indicating that there are non-negligible chances for the former to outperform the latter.
> > > > >
> > > > > We would also like to take the evaluation curves of CrazyClimber environment into a deeper discussion. It can be observed that the confidence interval of ***All-3-Step*** is wider than that of ***Mixed-1-3-Step***. As the confidence interval might not be sufficient to reflect on the real situation of the sampled data, we next provide another figure with all sampled data plotted instead of the confidence interval.
> > > > >
> > > > > > ***The figure for all sampled curves instead of the confidence interval is provided*** in the following link for the reviewer's reference: https://imgur.com/a/e6KGXCM
> > > > >
> > > > > The rationale behind the fact of the wider interval in ***All-3-Step*** is that, in two out of the five random seeds, the agent trained with ***All-3-Step*** setting fails and is unable to obtain any score. This results in large SEM. It implied that the agent trained with the ***All-3-Step*** setting is more unstable and thus leads to a wider interval. Although the upper bound of its interval is higher than that of ***Mixed-1-3-Step***, it is still not a good sign due to its unstability.
> > > > >
> > > > > We sincerely hope that the above explanation and the updated figures help for addressing the concerns of the reviewer.
> > > > >
> > > > > ---
> > > > >
> > > > > Q2. I think an interesting observation here is how when comparing to the heads of All-3-step, each head of Mixed-1-3-Step seems to do better and have less variability across the heads, which does seem in support of a better or more widely applicable representation being learned. I do like this table and think it better teases apart what is happening than the existing curves in the main text, assuming enough seeds are used to ensure statistical significance.
> > > > >
> > > > > A2. We would like to thank the reviewer for sharing the suggestions, insights, and opinions. We are also glad to have the chance to provide this additional table.  The table and the discussion will be included in the final version of the manuscript.

---

> > > ### Author Response · Authors · 2020-11-24
> > > **Response to Reviewer #2 (part 2/3)**
> > >
> > > Q3. Thank you for the additional description. ***I was more wondering whether it was something more formal, and perhaps measurable.*** While it has heads which implicitly represent different behaviors, I think the unclear part is why this would lead to better performance, as it's a little vague what is meant by benefitting from the "properties of different lengths of step returns." I feel the claim about additional exploration isn't very convincing due to DQN being generally off-policy, and ***one could take the supposedly less exploratory learner but provide a more exploratory behavior policy*** (e.g., ε-greedy with an annealing ε).
> > >
> > > ***Have the authors tried running MB-DQN (multiple heads with different backup lengths), but only behaving according to one of the heads (instead of only learning one head of some backup length)?*** From the differences observed with different fixed backup lengths, it seems possible that the differences may be due to predicting auxiliary tasks. That is, the varying multi-step update targets backpropagating through the same hidden layers may be having some regularization/representation learning effect.
> > >
> > > A3.
> > > We would like to thank the reviewer for raising this question, which is very interesting.  We are glad to share our thoughts and insights with the reviewer in the following paragraphs.
> > >
> > > Since we are not 100% sure about the exact meaning of the question raised by the reviewer, we would like to address it comprehensively in order to ensure that we will cover all the possible intentions of the question.
> > >
> > > To begin with, we hypothesize that the reviewer's question is whether the benefits of MB-DQN are mainly caused by the better representation learned through the multi-step bootstrapped heads.  In order to address this question, ***we would like to use the averaged score of each single head evaluated 10 times on the Seaquest environment in the following discussion.***  The scores for the bootstrapped heads (i.e., h1-h10) are presented in the following table.  For the ***Mixed-1-3-Step*** case, h1-h5 correspond to one-step backup length, while h6-h10 correspond to three-step backup length.  Please note that $\epsilon$-greedy is applied to MB-DQN as the original Bootstrapped DQN.
> > >
> > > ---
> > >
> > > |                | h1  | h2  | h3  | h4  | h5  | h6  | h7  | h8  | h9  | h10 | 1-Step Mean | 3-Step Mean | All-heads Mean | Voting Score |
> > > | :------------- | :----: | :----: | :----: | :----: | :----: | :----: | :----: | :----: | :----: | :----: | :-----------: | :-----------: | :--------------: | :------------: |
> > > | Mixed-1-3-Step | 3420 | 2251 | 3279 | 3626 | 2038 | 4408 | 4292 | 4472 | 4767 | 3528 | **2922**    | **4293**    | 3608           | 11395.0      |
> > > | All-3-Step     | 3716 | 2346 | 1012 | 2482 | 3464 | 3150 | 2600 | 4272 | 3300 | 1108 | NaN         | **2745**    | 2745           | 5470.0       |
> > >
> > > ---
> > >
> > > First, if the performance benefits are mainly due to the better representation learned through the multi-step bootstrapped heads, one can expect that the scores corresponding to different heads would be similar, as the heads share the same feature extraction network.  However, the results of the ***Mixed-1-3-Step*** case indicate that the performances of the one-step heads and the three-step heads differ.  Although the scores of the heads with the same backup length are similar (i.e., the h1-h5 group, or the h6-h10 group), the heads with different backup lengths lead to different scores of the agent.  This difference might imply that the behaviors of the agents corresponding to different heads are different.
> > >
> > > Second, for the performance of the three-step heads, the table reveals that the mean score of ***Mixed-1-3-Step*** (i.e., 4293) is larger than that of ***All-3-Step*** (i.e., 2745).  In addition, for the performance of the one-step heads, it is observed that the mean score of h1-h5 of ***Mixed-1-3-Step*** (i.e., 2922) is also larger than the mean score of all the heads of the ***All-3-Step*** case (i.e., 2745).  These might be partially due to the advantages from the better representation (as pointed out by the reviewer), and partially caused by the heterogeneity (as stated in the manuscript and our prior responses).
> > >
> > > Third, from the perspective of the performance gain from ***All heads mean*** to the ***Voting score***, it is observed that ***Mixed-1-3-Step*** is able to boost drastically from 3608 to 11395, while ***All-3-Step*** improves from 2745 to 5470.  This observation indicates that the heterogeneity in bootstrapped heads offers benefits when compared with the ***All-3-Step*** case.
> > >
> > > We hope that the above explanations can adequately address the reviewer's concerns.

---

> > > ### Author Response · Authors · 2020-11-24
> > > **Response to Reviewer #2 (part 1/3)**
> > >
> > > We sincerely appreciate the prompty reply from the reviewer, and would like to respond to the questions as follows.
> > >
> > > ---
> > > Q1. When following a head as a behavior policy, is it behaving deterministically greedy with respect to it? Many prior works would derive behavior from value estimates soft-greedily (e.g., ε-greedy, Boltzmann over Q) that it's still not clear how specifically the actions are being selected. I believe the observations about more exploratory behavior may be heavily intertwined with this design choice.
> > >
> > > A1.
> > > We would like to thank the reviewer for raising this question.  MB-DQN adopts the same action selection approach as the original Bootstrapped DQN [1].  During the training phase, ***MB-DQN uses an $\epsilon$-greedy policy***, in which the value of ***$\epsilon$ is annealed linearly from 1 to 0.01 over the first 1M timesteps***.  Please kindly note that the actions are not selected based on a deterministic greedy policy.
> > >
> > > We hope that the above clarification has adequately addressd the reviewer's question.
> > >
> > > [1] I. Osband *et al.* Deep exploration via Bootstrapped DQN.
> > >
> > > ---
> > >
> > > Q2. Can the authors comment on the updated results? I have looked at the album of results with the new 95% confidence intervals, and the statistical trends do not appear significant given the excessive overlap in the intervals.
> > >
> > > A2. We would like to thank the reviewer for raising this question and would like to elaborate on the results in terms of two aspects.  First, from the curves presented in the figure below, the trends of the mean scores (plotted in solid lines) of the three methods (i.e., ***Mixed-1-3-Step (MB-DQN)***, ***All-1-Step***, and ***All-3-Step***) show that the mean scores corresponding to MB-DQN are able to rise faster and/or higher than those of the other two baselines in most cases, and is comparable to them in Freeway. Second, from the perspective of the confidence intervals (CIs), it is observed that the CIs of the ***All-3-Step*** baseline are wider than those of ***Mixed-1-3-Step (MB-DQN)*** in several environments. From the above two perspectives, the statistical results indicate that the mixture usage of multi-step returns may offer benefits in mean scores as well as less variance than the baselines with longer backup lengths.
> > >
> > > > The experimental results for the Atari environments can be accessed at the following link for the reviewer's reference: https://imgur.com/a/FnzkuYy
> > >
> > > We hope that the above explanations can adequately address the reviewer's concerns.

---

> ### Author Response · Authors · 2020-11-21
> **Response to Reviewer #2 (part 1/2)**
>
> We sincerely thank the reviewer for spending time and providing valuable feedback. We would like to address the concerns from the reviewer by providing our responses.
>
>
> Q1:
> Could the authors elaborate on the behavior policy used while training, and comment on its compatibility with the use of uncorrected multi-step returns?
>
> A1:
> We are afraid that there might be some misunderstanding and would like to make a clarification to address the reviewer's concern.  ***Peng's Q($\lambda$) is used in neither the training phase nor the evaluation phase of MB-DQN.  Peng's Q($\lambda$) targets are only used by DQN($\lambda$)[1] and DQN($\lambda$) ensemble in Section 4.3*** for computing their $\lambda$-return targets, which are stored in memory cache proposed in [1].  Both DQN($\lambda$)[1] and DQN($\lambda$) ensemble only serve as the baseline methods to be compared with MB-DQN.
>
> We would like to clarify that at the beginning of each episode, the agent picks up a random bootstrapped head as the behavior policy. It then collects experiences using that policy during the same episode, and the collected experiences are store in the replay buffer, which is shared by all of the bootstrapped heads. As a result, the update target is not a mixture of on-policy sampled rewards. Each head has its own return target with a fixed backup length.  Neither correction nor modification is applied to the returns collected by the agent, which is the same as the multi-step approach adopted in RainbowDQN [2].  The training process of MB-DQN is the same as that of the original Bootstrapped DQN for a fair comparison and is described in Section 3.2.
>
> We hope that the above clarification can address the question from the reviewer.  Please kindly let us know if there is any further question on this part.
>
> [1] B. Daley and C. Amato. Reconciling $\lambda$– returns with experience replay.
>
> [2] M. Hessel *et al.* Rainbow: Combining improvements in deep reinforcement learning.
>
> ---
>
> Q2. When looking at the heatmaps after 100,000 time steps, the 1-step learner has a very salient line taking a suboptimal path of going straight up for a bit and then heading in a straight line toward the goal. Such a high frequency of this suboptimal path seems suggestive of the method actually not exploring enough.
>
> A2.
> We would like to thank the reviewer for pointing out this issue and would like to apologize for misleading the reviewer.
>
> We would like to clarify that the trajectory of the agent in an episode is "optimal" as long as it reaches the target and is composed of "moving up" and "moving right" actions.  The rationale is that the environment is a grid-world, in which all such trajectories contain the same number of moves.  As a result, the path preferred by the one-step learner at 100,000 timesteps is actually optimal and has the same number of moves as the five-step learner.
>
> However, we would also like to clarify that the five-step learner converges to the optimal path faster than the one-step learner.  It can be observed that at 50,000 timesteps, the five-step learner has already learned an optimal path to follow, while the one-step learner is still exploring the environment.  This observation validates that different numbers of backup lengths lead to different behavior policies of the agents.
>
> ---
>
> Q3. Only 3 random seeds were used in their empirical evaluation, and the paper presents one standard deviation as a 68% confidence interval.  ***Can the authors comment on the statistical significance of their results, and whether the number of runs is sufficient for the claims being made?***
>
> A3. We appreciate the suggestion from the reviewer.  We have raised the number of random seeds of MB-DQN and the All-3-Step agents to five and modify the confidence interval to 95%.  The statistical trends of the updated curves for MB-DQN and the baselines still support the findings of this research. We will be glad to further increase the number of runs in the final version of the manuscript.
>
> > ***We provide the additional experimental results with 5 seeds and modify the confidence interval to 95%*** in the following link: https://imgur.com/a/FnzkuYy

---

### Official Review · AnonReviewer3 · 2020-10-30
**Interesting concept, well written paper. More experiements would have been more convincing**

**Rating:** 7
**Confidence:** 3

**Review:**

Summary:
As the title suggests, the paper proposes using a mixture of multi-steps in estimating the action-space value function in DRL. Multi-step RL has been around for a while now, and this work attempts to extend it further to a mixture setting by exhibiting a performance boost when the proposed technique is used with Bootstrapped DQN.
Why multi-steps?
Multi-step return estimations constitute a high-variance low-bias method. Because of the “intuition” that different multi-steps would incorporate different knowledge, and the step parameter might be dependent on the environment dynamics, it would make sense to have several of them.
Why Bootstrapped DQN:
Provides easy incorporation of the multi-steps since it has multiple Q heads, and the authors could assign different multi-steps to them.

The paper is easy to read and follow. The different sections have a “natural” flow to them and the reasoning is clear.
To prove the advantage of using the algorithm, it has been tested on 8 of the Atari games and it exhibited better performance in several of them. A qualitative comparison was also performed based on the attention map visualization for 2 Atari games and it showed that MB-DQN infers more relevant sate-space regions.

“However” points:
The critical point is that the performance of the baseline Bootstrapped DQN is different from the one reported in the original paper. We would assume there is still some hyperparameter tuning missing for B-DQN. Yet, this does not disapprove of the power of the suggested MB-DQN since the same hyperparameters were used for both methods.
Since MB-DQN is expected to have a higher variance, it would be interesting to test in a noisy environment (non-deterministic action-reward?).
(Slight indentation problem in the first hyperparameters table that pushed the head return value to the first row)

Verdict:
This is an interesting and powerful method. Well written paper. More extensive experiments would have made the proposed work more convincing.

---

> ### Author Response · Authors · 2020-11-21
> **Response to Reviewer #3**
>
> We sincerely thank the reviewer for spending time and providing valuable feedback. We would like to address the concerns of the reviewer by providing our responses.
>
> Q. “However” points: The critical point is that the performance of the baseline Bootstrapped DQN is different from the one reported in the original paper. We would assume there is still some hyperparameter tuning missing for B-DQN. Yet, this does not disapprove of the power of the suggested MB-DQN since the same hyperparameters were used for both methods. Since MB-DQN is expected to have a higher variance, it would be interesting to test in a noisy environment (non-deterministic action-reward?). (Slight indentation problem in the first hyperparameters table that pushed the head return value to the first row)
>
> A. We would like to sincerely thank the reviewer, would also like to bring to the reviewer's kind attention that the curves presented in the original paper of Bootstrapped DQN are presented in terms of frames, while those in our manuscript are presented in terms of timesteps. For each timestep, four frames are used. In other words, the experiments in the original paper of Bootstrapped DQN are performed for 50M timesteps, which are five times than ours. The baseline results presented in our paper are actually very similar to those reported in the original paper.
>
> We would also like to thank the reviewer for pointing out the table formatting issue. We will fix it and update the manuscript in the final version.
>
> We agree with the reviewer that the investigation of the environments with non-deterministic action-rewards is an interesting topic, and will be a direction of our future research.

---

### Official Review · AnonReviewer5 · 2020-11-07
**MB-DQN**

**Rating:** 5
**Confidence:** 4

**Review:**

Summary:

This paper proposes Mixture Bootstrapped DQN (MB-DQN). The new algorithm modifies bootstrapped DQN by giving different backup lengths to each head. It shows better performance in several Atari environments, compared to bootstrapped DQN. This work also attempts to analyze the source of empirical benefits in terms of attention areas of the agents, the quality of data sampling, and the way of utilizing different backup lengths. There is also an ablation analysis showing that the MB-DQN is not limited to a single configuration.

Reasons for score:

(+): The proposed method is tested considering several different aspects, including the agent’s attention area, data quality, and the sensitivity of parameters. These experiments provide a deeper view of the MB-DQN and try to explain why this new method helps with learning.

(+): The paper is well organized and easy to read. The appendix provides enough details to reproduce the main result.

(-): This paper claims that while longer back-ups expedite learning, they do so at the expense of exploration: longer back-ups help in efficient assimilation of the reward information that the agent begins to exploit, as opposed to shorter TD(0)-like backups that propagate reward information relatively slowly, leading to seemingly exploratory behavior (reward information has not propagated everywhere so exploitation essentially amounts picking random actions). I find this assertion rather flawed as it suggests that inefficient learning is an avenue for better exploration. An agent tends to explore when it chooses a seemingly sub-optimal course of action to improve its knowledge, and it tends to exploit its current knowledge when it chooses the greedy action. Deliberately slowing the learning process such that even exploitation appears exploratory is a defeatist approach towards the exploration problem.

(-): While there is some empirical comparison with eligibility traces in section 4.3, the fundamental advantages of using multiple heads over TD-lambda remains unclear to me. Can we see the benefits clearly in the tabular/linear function approximation setting, or make a theoretical statement about the benefits? Or is this approach only useful in the Deep-RL setting where it helps with how neural networks interact with TD? Section 4.2 suggests that MB-DQN helps by changing the data distribution in the replay buffer, can we see the same effect in the tabular/linear function approximation setting, perhaps in the domain used in Figure 1? Studying these questions empirically in simpler settings can help untangle the interaction of NN-based non-linear function approximation and the general temporal credit assignment problem that traces are designed to solve.

(-): The experimental results are averaged over only 3 different random seeds. It remains unclear how these seeds were chosen. More importantly, 3 runs are not sufficient to establish the statistical significance of the results.

---

> ### Author Response · Authors · 2020-11-21
> **Response to Reviewer #5 (part 2/2)**
>
> Q3. The experimental results are averaged over only 3 different random seeds. It remains unclear how these seeds were chosen. More importantly, 3 runs are not sufficient to establish the statistical significance of the results.
>
> A3. Many thanks for the suggestion.  ***We have raised the number of random seeds of MB-DQN and the All-3-Step agents to five***, which is a commonly adopted number used by the RL literature [1, 2], and illustrate the new comparisons in the following figure.
>
> > ***We provide the additional experimental results with 5 seeds*** in the following link: https://imgur.com/a/FnzkuYy
>
> Please kindly note that the seeds used in this research are randomly chosen (as mentioned in Footnote 1. in our manuscript), and the statistical trends of the updated curves for MB-DQN and the baselines still support the findings of this research.  We will be glad to further increase the number of runs in the final version of the manuscript.
>
> ---
>
> Response to the concerns of R1
>
> Q4. I find this assertion rather flawed as it suggests that inefficient learning is an avenue for better exploration. An agent tends to explore when it chooses a seemingly sub-optimal course of action to improve its knowledge, and it tends to exploit its current knowledge when it chooses the greedy action. Deliberately slowing the learning process such that even exploitation appears exploratory is a defeatist approach towards the exploration problem.
>
> A4. We sincerely thank the reviewer for raising this question.  We would like to feedback and respectfully express our concerns in terms of two aspects.  First, we feel that this question seems to involve the fundamental dilemma regarding the purpose and design of the exploration technique, which can be either undirected [2-7] and directed [8-11].  There have been a number of previous research works dedicated to both of them, and have been demonstrated effectively.  Although incorporating directed exploration techniques may further enhance the performance, MB-DQN adopts the same exploration concept same as Bootstrapped DQN [12] that does not involve directed exploration for the purpose of a fair comparison.
>
> Second, we would like to bring to the reviewer's kind attention that allowing the agent to collect a diverse set of experiences (even with sub-optimal trajectories) is beneficial to the learning process of its behavioral policy [2-7], and has also been demonstrated in the experimental results in Section 4.2 and Fig. 5(a).  This paper highlights the importance of different backup lengths in multiple bootstrapped heads, and show that the proposed approach is more effective than the previous TD ($\lambda$) based methods [13,14] that combine the target returns to a single one.
>
> We hope that the above explanation addresses the reviewer's concerns.
>
> ---
> ### Reference
>
> [1] Y. Duan *et al.* Benchmarking Deep Reinforcement Learning for Continuous Control
>
> [2] V. Mnih *et al.* Asynchronous methods for deep reinforcement learning
>
> [3] V. Mnih *et al.* Human-level control through deep reinforcement learning.
>
> [4] T. Haarnoja *et al.* Soft Actor-Critic: Off-Policy Maximum Entropy Deep Reinforcement Learning with a Stochastic Actor
>
> [5] M. Plappert *et al.* Parameter Space Noise for Exploration
>
> [6] M. Fortunato *et al.* Noisy Networks for Exploration
>
> [7] T. P. Lillicrap *et al.* Continuous control with deep reinforcement learning
>
> [8] D. Pathak *et al.* Curiosity-driven exploration by self-supervised prediction.
>
> [9] Y. Burda *et al.* Large-Scale Study of Curiosity-Driven Learning
>
> [10] Y. Burda *et al.* Exploration by random network distillation.
>
> [11] Z. Hong *et al.* Diversity-Driven Exploration Strategy for Deep Reinforcement Learning
>
> [12] I. Osband *et al.* Deep exploration via bootstrapped dqn.
>
> [13] B. Daley and C. Amato. Reconciling $\lambda$–eturns with experience replay
>
> [14] R. S. Sutton and A. G. Barto. Reinforcement Learning: An Introduction.

---

> > ### Comment · AnonReviewer5 · 2020-11-23
> > **Reply**
> >
> > Thank you for your detailed reply and for adding more experiments.
> > The idea of separating the data sampling and value function learning steps is nice.
> >
> > A2.
> > I appreciate the detailed explanations. Though MB-DQN holds multiple backup-lengths, it needs to do leverage in control---MB-DQN uniformly and randomly selects a head (according to section 3.2). That seems like a different weighting on backup lengths, is that right or there are other things I am missing here? If it is a different weighting, how does this weighting provide an advantage compared to the lambda return? Does it suggest that it may not be necessary to reduce the weight as the backup length becomes longer?
> >
> > A4.
> > Thanks for pointing out the definition in paper [12] and explaining these in detail, though I think [12] also mentioned that the exploration includes "sacrificing immediate rewards". Maybe adding a similar environment as used in [12], to demonstrate the exploration advantage of MB-DQN, would be more convincing. (Like the environment in Figure 3 in [12], with a little bit tricky reward setting that requires the agent to explore for a larger reward after it has learned a smaller reward.)

---

> > > ### Author Response · Authors · 2020-11-24
> > > **Response to Reviewer #5**
> > >
> > > We sincerely appreciate the prompt reply from the reviewer and would like to respond to the questions as follows.
> > >
> > > ---
> > >
> > > Q1. I appreciate the detailed explanations. Though MB-DQN holds multiple backup-lengths, it needs to do leverage in control --- MB-DQN uniformly and randomly selects a head (according to section 3.2). ***That seems like a different weighting on backup lengths, is that right or there are other things I am missing here?*** If it is a different weighting, how does this weighting provide an advantage compared to the lambda return? ***Does it suggest that it may not be necessary to reduce the weight as the backup length becomes longer?***
> > >
> > > A1.
> > > We would like to thank the reviewer for raising this question.  In order to address the question, we would like to bring to the reviewer's kind attention that MB-DQN does not apply weights on different return targets, and thus is different from TD($\lambda$) and DQN($\lambda$). Each head has its own return target with a fixed backup length. At the beginning of each episode, the agent picks up a random bootstrapped head for determining its actions. It then collects experiences using that policy during the same episode. The collected experiences are stored in the replay buffer, which is shared by all of the bootstrapped heads.
> > >
> > > During the updating phase, each head samples transitions from the shared replay buffer, and calculates the target value based on its own backup length. As a result, MB-DQN does not rely on weighing different return targets as TD($\lambda$) and DQN($\lambda$).  Please note that the target function of MB-DQN still employs the discount factor $\gamma$ when calculating the n-step return for each head, as stated in Section 3.2 and Algorithm 1 of the original manuscript.
> > >
> > > We hope that the above explanations can adequately address the reviewer's concerns.
> > >
> > > ---
> > >
> > > Q2. Thanks for pointing out the definition in paper [12] and explaining these in detail, though I think [12] also mentioned that the exploration includes "sacrificing immediate rewards". Maybe adding a similar environment as used in [12], to demonstrate the exploration advantage of MB-DQN, would be more convincing. (Like the environment in Figure 3 in [12], with a little bit tricky reward setting that requires the agent to explore for a larger reward after it has learned a smaller reward.)
> > >
> > > A2. We would like to thank the reviewer for the suggestion.  In our paper, we use the grid-world maze as our motivational example, as it provides an easier way for visualizing and comparing the visited states of the baseline approaches with different backup lengths.  The environment used in [12] will be one of our considerations for demonstrating the exploration behaviors in future work.

---

> > > > ### Comment · AnonReviewer5 · 2020-11-24
> > > > **Reply**
> > > >
> > > > Thank you for the detailed reply to my concerns. The concern about random seed, data distribution, and the difference between MB-DQN and DQN(lambda) have been addressed. I would still like to see experimental results showing the advantage of the method in the simpler tabular/linear FA setting, possibly in the same grid-world example.

---

> > > > > ### Author Response · Authors · 2020-11-25
> > > > > **Response to Reviewer #5**
> > > > >
> > > > > Q1. Thank you for the detailed reply to my concerns. The concern about random seed, data distribution, and the difference between MB-DQN and DQN(lambda) have been addressed. I would still like to see experimental results showing the advantage of the method in the simpler tabular/linear FA setting, possibly in the same grid-world example.
> > > > >
> > > > > A1. We would like to thank the reviewer for raising this question.  To address the concerns from the reviewer, we use the same grid-world maze environment and perform experiments for the linear function approximator versions of ***Mixed-1-5-Step***, ***All-1-Step***, and ***All-5-Step***, as suggested by the reviewer. The results of the linear function approximator versions are provided in the following figure.
> > > > >
> > > > > > The state visitations of the linear function approximator versions are provided in the following link for the reviewer's reference: https://imgur.com/a/Wc43dWe
> > > > >
> > > > > From the results, it is observed that for the linear function approximator versions, the ***All-1-Step*** case still tends to explore the environment, while the ***All-5-Step*** case still quickly converges to a path to the goal.  The ***Mixed-1-5-Step*** case explores the environment as well, but not converges as fast as the ***All-5-Step*** case.  Please kindly note that the state visitations of the linear function approximator versions differ from those of the DRL versions, as the designs of the function approximators differ.
> > > > >
> > > > > We hope that the above new experimental results would help to address the concerns of the reviewer.

---

> ### Author Response · Authors · 2020-11-21
> **Response to Reviewer #5 (part 1/2)**
>
> We sincerely thank the reviewer for spending time and providing valuable feedback.  We would like to address the concerns from the reviewer by providing our responses as well as our additional experimental results.
>
> ---
>
> Q1. Section 4.2 suggests that MB-DQN helps by changing the data distribution in the replay buffer, ***can we see the same effect in the tabular/linear function approximation setting, perhaps in the domain used in Figure 1***?
>
> A1. Thanks for raising this question.  As suggested by the reviewer, we are glad to leverage the grid-world setting shown in Fig. 1 of our manuscript to demonstrate the impact of MB-DQN on data distribution.
>
> > ***We provide the additional experimental results*** in the following link: https://imgur.com/a/FqnndK0
>
> In this figure, ***we compare the states visited by (a) All-1-Step, (b) Mixed-1-5-agent, and (c) All-5-Steps, among which (b) corresponds to MB-DQN.*** The states visited by the agents can be viewed as the data distribution of their experience. It can be observed that the data distribution of Mixed-1-5-agent (i.e., MB-DQN) possesses the characteristics of both All-1-Step and All-5-Step, where the former is more diverse and the latter is more concentrated.  Although there is no straightforward way to directly compare the data distributions of the cases presented in Fig. 5(a) of Section 4.2 due to the high-dimensional inputs of Atari environments, we believe that the above motivational example can serve as an interpretation for the impacts of MB-DQN on data distribution, which is more diverse and is benefited from different bootstrapped lengths.  In addition, the relatively higher scores of MB-DQN in Fig. 5(a) also reflects the benefits from this characteristic.
>
> ---
>
> Q2. The fundamental advantages of using multiple heads over TD-lambda remains unclear to me. ***Can we see the benefits clearly in the tabular/linear function approximation setting, or make a theoretical statement about the benefits?*** Or is this approach only useful in the Deep-RL setting where it helps with how neural networks interact with TD?
>
> A2. We appreciate the reviewer for raising the question.
>
> In order to address the question from the reviewer, we would like to use the two-dimensional grid world maze environment same as the previous question to ***demonstrate the experimental results for the DQN ($\lambda$) setting shown in the (d) column in the following figure.***
>
> > ***We provide the additional experimental results*** in the following link: https://imgur.com/a/FqnndK0
>
> It can be observed that the agent trained with $\lambda$-return converges faster, and reaches the goal through a more concentrated path. The rationale behind this observation might be due to the long horizon of the backup lengths used by DQN ($\lambda$) for deriving the target value. However, it is also observed that the agent trained with $\lambda$-return explores apparently fewer states than those visited by the All-1-Step, All-5-Step, and Mixed-1-5-Step agents. The results on this grid world environment also validate the advantages of using multiple heads over TD ($\lambda$), as MB-DQN enables heterogeneity in return targets.
>
> In order to further elaborate on the above observations, we would like to separately explain the differences between TD ($\lambda$) based methods and MB-DQN. TD ($\lambda$) based methods have long been used as a conventional approach for combining the step returns from different backup lengths [1].  More recently, DQN ($\lambda$) proposed a specially designed replay buffer and cache to integrate the concept of TD($\lambda$) to resolve the difficulty of applying TD ($\lambda$) to deep reinforcement settings.  This is because TD ($\lambda$) requires a set of records of states and rewards so as to derive the ($\lambda$) returns with different backup lengths.  These approaches (either TD ($\lambda$) or DQN ($\lambda$)) integrates different return targets by combining them into a single unified target, which may potentially sacrifice advantages from the heterogeneity in different return targets.
>
> On the other hand, our objective and motivation are to embrace a new way of combining different step returns through multiple bootstrapped heads.  Multiple bootstrapped heads allow MB-DQN to leverage the advantages from different return targets, enabling the agents to learn different behaviors through the use of multiple return targets instead of a single one.  As a result, MB-DQN exhibits better performance than DQN ($\lambda$) in the experiments presented in Fig. 5(b) and Section 4.3.
>
> We hope that the above explanations have adequately addressed the reviewer's questions.
>
> [1] R. S. Sutton and A. G. Barto. Reinforcement Learning: An Introduction.

---

### Decision · Program_Chairs · 2021-01-07
**Final Decision**

**Decision:**

Reject

**Comment:**

This paper extends Bootstrap DQN with multi-step TD target. The initial submission had missing details, communication problems, and results lacking rigor. The authors made a clear effort to address the reviewers concerns.

This paper's contribution is supported primarily by the empirical results which need major work. The lack of statistical significance in the key results is a major problem. The new 5 run results (originally only 3 runs) shows no clear evidence of improving over the baseline. Additionally one must either justify the use of such few runs by investigating the distributions and using the proper statistical tools (Colas et al [2]) or simply do more runs. Regardless statistical significance in the precise sense is a requirement.

In addition other adjustments to the paper would strengthen it significantly: (1) The qualitative results like state visitations can be interpreted either in favour of the method or not, this could be improved with discussion or omitted---see [1]; (2) the discussion of heterogeneity was informal; (3) discussion of the impact and sensitivity of hyper-parameters should be included---this includes addressing the concern that the performance of the baseline was as strong as it could be; (4) the current results do clearly separate if the improvement in performance (if it can be shown to be significant) is due to improvements in the rep via auxiliary task effect or the multi-step return---the reviewer has made a nice suggestion for an experiment here.

In summary, the reviewers did not find the text and examples in the paper convincing as to why the proposed method should be better than bootstrap q, and the results are not significant and need more work.

references that may be helpful:
[1] https://openreview.net/forum?id=rkl3m1BFDB&utm_campaign=RL%20Weekly&utm_medium=email&utm_source=Revue%20newsletter
[2] https://arxiv.org/abs/1806.08295